# PhotoFiTT: a quantitative framework for assessing phototoxicity in live-cell microscopy experiments

Mario Del Rosario [1,2,10], Estibaliz Gómez-de-Mariscal [1,2,3,10], Leonor Morgado[1,2,3,4], Raquel Portela[1], Guillaume Jacquemet [5,6,7,8], Pedro M. Pereira [1] ✉ & Ricardo Henriques [1,2,9] ✉

Phototoxicity in live-cell fluorescence microscopy can compromise experimental outcomes, yet quantitative methods to assess its impact remain limited. Here, we present PhotoFiTT (Phototoxicity Fitness Time Trial), an integrated framework combining a standardised experimental protocol with advanced image analysis to quantify light-induced cellular stress in label-free settings. PhotoFiTT leverages machine learning and cell cycle dynamics to analyse mitotic timing, cell size changes, and overall cellular activity in response to controlled light exposure. Using adherent mammalian cells, we demonstrate PhotoFiTT's ability to detect wavelength- and dose-dependent effects, showcasing that near-UV light induces significant mitotic delays at doses as low as 0.6 J/cm$^2$, while longer wavelengths require higher doses for comparable deleterious effects. PhotoFiTT enables researchers to establish quantitative benchmarks for acceptable levels of photodamage, facilitating the optimisation of imaging protocols that balance image quality with sample health.

Live-cell fluorescence microscopy has revolutionised our ability to study dynamic biological processes in near-native conditions[1]. However, the intense illumination required can induce phototoxicity, disrupting cellular functions[2–5]. These effects, often subtle and cumulative, can lead to significant changes in cell behaviour, organelle integrity, and developmental processes[6–10]. Yet, establishing optimal imaging protocols that balance high-quality data acquisition with minimal biological interference remains a complex challenge[11]. Traditional methods for assessing phototoxicity include viability assays and morphological observations[2,3]. Historically, photobleaching has been used as a proxy reporter of phototoxicity[2,5,12], but with the advent of highly photostable fluorescent proteins like mStayGold[13], this approach has become less reliable. Importantly, specimen type, developmental stage, and experimental conditions influence phototoxicity tolerance, complicating the development of replicable imaging protocols[14,15]. Previous studies have offered valuable quantitative assessments of phototoxicity[3,10,12,14,16–19], including the use of advanced deep-learning algorithms as classifiers[20] and mitosis analysis[21]. Yet, a critical need remains for standardised and generalisable methods able to quantitatively link cell damage to high-irradiance light exposure across different fluorescence microscopy techniques. Addressing this gap, we introduce PhotoFiTT (Phototoxicity Fitness Time Trial), a quantitative imaging-based framework designed to assess phototoxicity effects on cellular behaviour in live-cell microscopy

[1]Instituto de Tecnologia Química e Biológica António Xavier, Universidade Nova de Lisboa, Oeiras, Portugal. [2]Instituto Gulbenkian de Ciência, Oeiras, Portugal. [3]Gulbenkian Institute for Molecular Medicine, Oeiras, Portugal. [4]Abbelight, Cachan, France. [5]Turku Bioimaging, University of Turku and Åbo Akademi University, Turku, Finland. [6]Faculty of Science and Engineering, Cell Biology, Åbo Akademi University, Turku, Finland. [7]InFLAMES Research Flagship Center, Åbo Akademi University, Turku, Finland. [8]Turku Bioscience Centre, University of Turku and Åbo Akademi University, Turku, Finland. [9]UCL Laboratory for Molecular Cell Biology, University College London, London, United Kingdom. [10]These authors contributed equally: Mario Del Rosario, Estibaliz Gómez-de-Mariscal. ✉e-mail: pmatos@itqb.unl.pt; r.henriques@itqb.unl.pt

experiments (Fig. 1) for adherent cell lines in traditional coverslip-based setups. Here, we show how PhotoFiTT provides a rigorous, label-free, and quantitative approach to evaluate phototoxicity, enhancing the reliability and reproducibility of live-cell imaging studies. It is designed as an integrated framework comprising a standardised experimental protocol and an advanced image analysis pipeline. PhotoFiTT leverages the predictable nature of cell division[22] as a 'biological clock' to quantify phototoxicity-induced perturbations (Fig. 1). The framework uses low-irradiance illumination brightfield microscopy to monitor cell populations that have been exposed to controlled light damage events. It employs deep-learning algorithms, including virtual staining and cell segmentation, to aid in tracking abnormal light-induced cellular behaviour. PhotoFiTT employs three key measurements: (1) Mitosis monitoring: assess delays in cell division by mapping cell rounding in time (Fig. 1a); (2) Cell size dynamics: track changes in cell size to discriminate between cell division and cell cycle arrest (Fig. 1b); (3) Cellular activity: quantify overall cellular changes over time as a measure of overall cellular health (Fig. 1c). Cellular activity reflects general dynamic changes in cellular shape across time. By analysing these factors, PhotoFiTT enables researchers to extract numerical constraints for optimising live-cell-compatible illumination conditions.

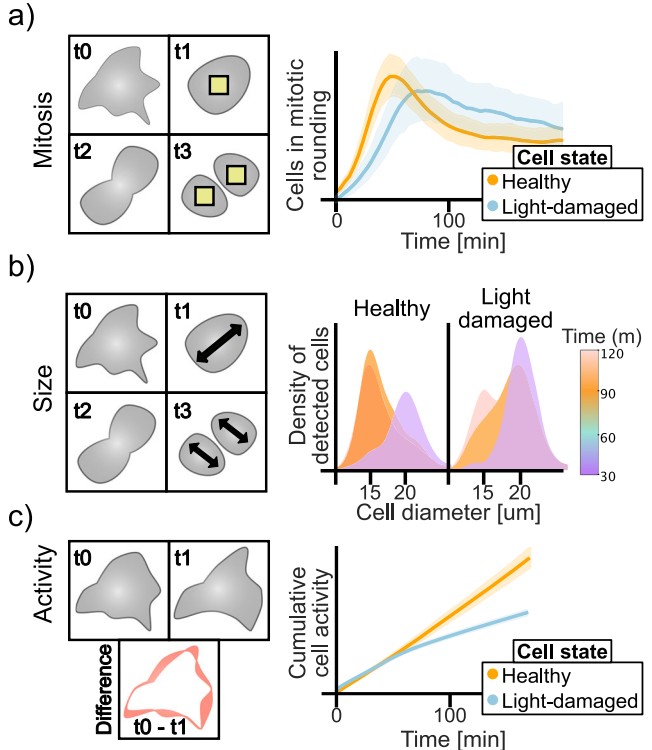

**Fig. 1 | Phototoxicity fitness time trial (PhotoFiTT) assay.** PhotoFiTT integrates a experimental imaging assay with an image analysis workflow, using cellular processes and mitotic cycles as natural timers to track consistent patterns of cell behaviour. It detects deviations in these patterns caused by light exposure, enabling the quantification of phototoxic effects in fluorescence microscopy. This is only the case where a dependence on light dose (or irradiance) has been demonstrated. The patterns can also be caused by other factors (temperature, media, cumulative damage, etc.) and thus, comparison against a control is also recommended. The workflow analyses three cellular features: **a** Mitosis: Identifies cell rounding (yellow squares), typically 30–50 min post-G2 exit. Phototoxicity alters rounding time distributions (orange: normal, blue: photodamaged). **b** Size: Tracks cell diameters (black arrowheads) through division. Mother cells (purple) transition to daughter cells (orange) ~60 min post-G2 exit. Phototoxicity delays this transition (light orange). **c** Activity: Quantifies observable cellular changes between frames (red outlines). Here cumulative cell activity represents the sum of activity in time. Normal cells show increased post-division activity (orange, around 60 min), while photodamaged cells (blue) exhibit reduced activity.

## Results

In this study, we applied the PhotoFiTT concept using synchronised cell populations in a simplified framework. This approach allowed us to examine multiple mitotic events under highly controlled conditions. We also measured unsynchronised populations as controls to identify any anomalies that might arise from the synchronisation process and its possible influence on the phototoxicity sensitivity of the examined cell populations[23]. This implementation (Fig. 2a), followed these key steps: (1) Synchronise cells using the CDK1 inhibitor RO-3306[24], which arrests cells in the G2/M interface, or study unsynchronised cells that can be tracked from cell rounding until division; (2) Expose cells to an illumination event replicating the pattern of the imaging experiment to be analysed for phototoxicity; (3) Identify undergoing cell rounding and division to separate mitotic rounding from stress-induced cell rounding (Fig. 2b) using a Stardist[25] deep-learning model purposely-trained for brightfield round cell detection; (4) Analyse three key parameters: mitotic timing, cell diameter dynamics, and cellular activity. Supplementary Movies 1 and 2 provide in-detail tutorials to reproduce these procedures. We conducted a comprehensive analysis of phototoxicity primarily using Chinese Hamster Ovary (CHO) cells as a model system and further tested on HeLa cells. CHO cells have the benefit of having a short doubling time (14–17 h), allowing for efficient experimental timelines while being a highly popular cell line for microscopy and cell research. Both synchronised and unsynchronised cell populations were exposed to varying doses of near-UV (385 nm), blue (475 nm) and red (630 nm) light to assess wavelength-dependent effects. In synchronised cells and in agreement with established literature[18,26], near-UV light dose induced significant dose-dependent delays in mitotic timing, with effects detectable at doses as low as 0.6 J/cm² (Fig. 2c, d and Supplementary Movie 3). Notably, at 60 J/cm², the characteristic peak in mitotic rounding became almost imperceptible, indicating widespread cell cycle arrest due to stress-induced cell rounding. These light doses are comparable to standard widefield microscopy imaging (Table S1). To validate that these effects were primarily due to light exposure rather than the potential light sensitivity of the CDK1 inhibitor used for synchronisation, we conducted parallel experiments with unsynchronised cells and manually tracked cell rounding and mitosis. In unsynchronised cell populations exposed to near-UV light, we observed a dose-dependent decrease in dividing cells (an 80% decrease in mitotic events on light doses of 60 J/cm²) and a simultaneous increase in arrested cells (a 30% increase of arrested cells on light doses of 60 J/cm²) (Fig. S1a). Consistent with our findings in synchronised populations, the time required for cells to divide after entering mitotic rounding increased with higher light doses from an expected 30 min division time to over 55 min in the case of 60 J/cm² (Fig. S1b). These results confirm that the observed effects were primarily attributable to light exposure rather than light sensitivity increase by the synchronisation drug. Notably, synchronised populations exhibited higher sensitivity to near-UV-induced damage compared to their unsynchronised counterparts (2). This increased susceptibility was evidenced by a higher proportion of apoptotic cells, as identified by a SYTOX-based cell viability assay (Fig. S1c), where increasing the excitation light dose led to an increase of up to fivefold in the number of dead cells. This observation can be attributed to the fact that synchronised cells were uniformly arrested at the G2/M checkpoint, a critical stage immediately following DNA replication and preceding mitosis. Near-UV radiation is known to induce DNA damage[14], and cells in G2 phase are particularly vulnerable due to the presence of fully replicated chromosomes and the imminent onset of mitosis. The accumulation of DNA damage at this stage can trigger cell cycle arrest or apoptosis more readily than in unsynchronised populations, where cells are distributed across various cell cycle phases[27]. Additionally, as a control, we manually tracked cell rounding and mitosis in synchronised CHO cell populations to compare against our automated detection pipeline. The results followed the same trend of a

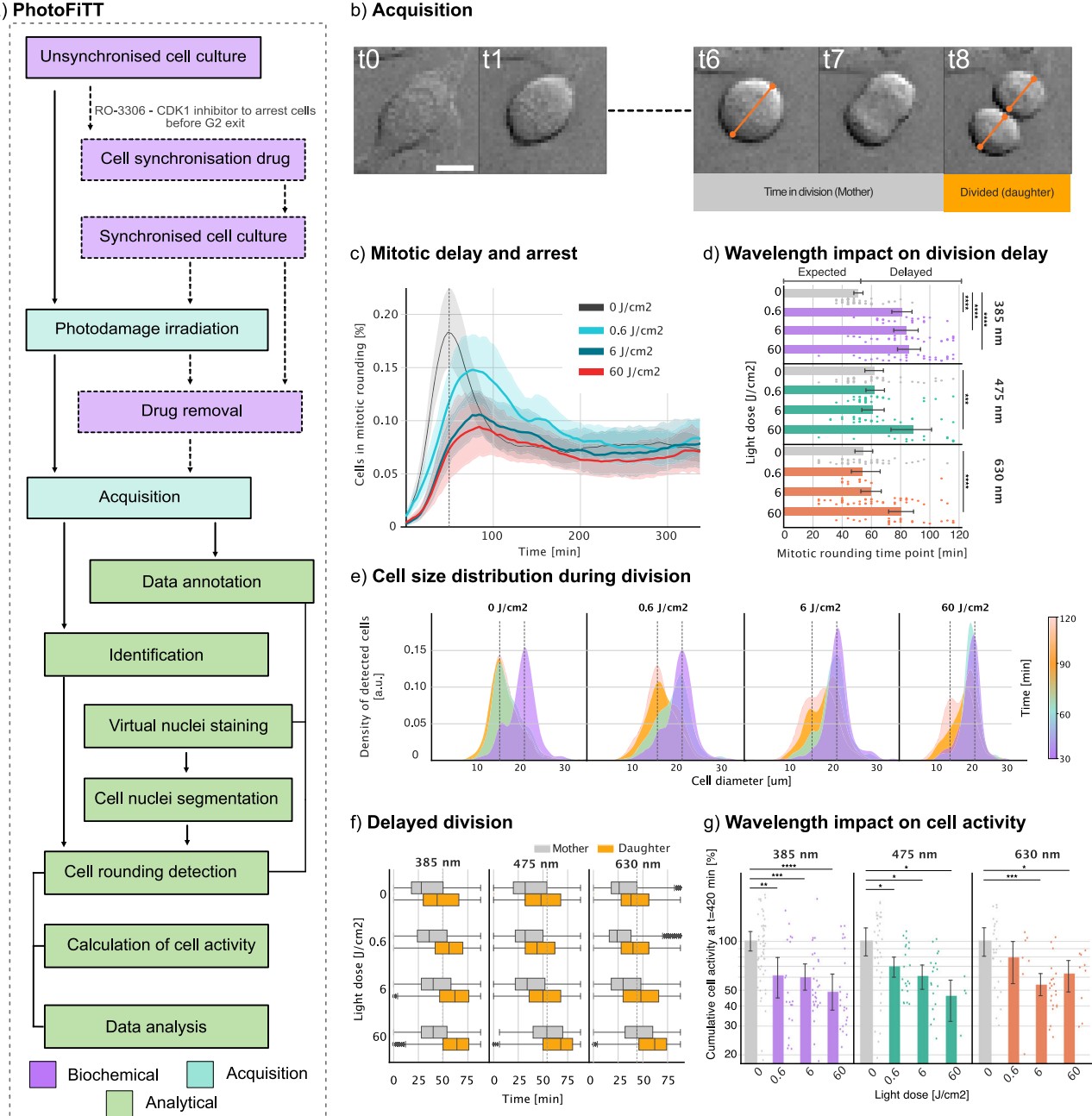

**Fig. 2 | Quantitative phototoxicity assessment with PhotoFiTT. a** PhotoFiTT integrates biochemical experiments, microscopy acquisition, and image analysis. Cell synchronisation is induced by CDK1 inhibition (RO-3306), arresting cells at the G2/M interface. Synchronised cells are washed prior to 6-h time-lapse imaging, including transmitted light and fluorescence markers for model training. Cell rounding is identified using trained models to quantify cell activity and division dynamics. Detailed workflow descriptions are in Notes S1 and S2. **b** Time-lapse brightfield imaging of an adherent mammalian cell (CHO Chinese hamster ovary) undergoing mitosis. The mother cell's diameter approximately doubles that of the resulting daughter cells post-division. Scale bar 20 μm. **c** Temporal distribution of mitotic cell rounding in synchronised populations exposed to varying doses of 385 nm (near-UV) light J/cm². Exposed populations exhibit a dose-dependent delay in cell rounding, manifested as rightward shifts in the distribution peaks. **d** Quantification of cell rounding delays across different light wavelengths and doses. The control population peaks at 50 min, representing the average cell rounding time (indicated with a line in **b**). Exposure to 385 nm light induces a dose-dependent delay, while 475 and 630 nm wavelengths only cause significant delays at high doses (60 J/cm²). **e** Cell volume nearly halves as mother cells divide into two daughter cells. Low-dose exposure (0.6 J/cm²) delays this transition, while high doses result in heterogeneous populations at later time points (120 min), indicating

asynchronous divisions and potential cell cycle arrest. **f** Temporal analysis of mother and daughter cell populations across different wavelengths and doses. All wavelengths induce delays in daughter cell appearance, with 385 nm light and high doses of 475 and 630 nm light causing the most pronounced effects. **g** Cumulative cell activity over 7 h post-exposure normalised for each replica (logarithmic scale). All light exposures reduce overall cell activity, with 385 and 475 nm showing more dose-dependent effects than 630 nm light. These results collectively demonstrate the wavelength- and dose-dependent impacts of light exposure on cell division dynamics and overall cellular activity. *N* values correspond to each acquired field of view. In 385 nm (0 (*n* = 130), 0.6 (*n* = 44), 6 (*n* = 22), and 60 (*n* = 34)), 475 nm (0 (*n* = 80), 0.6 (*n* = 25), 6 (*n* = 15), and 60 (*n* = 5)), and 630 nm (0 (*n* = 57), 0.6 (*n* = 39), 6 (*n* = 38), and 60 (*n* = 16)). A total of five biological replicates with ten technical replicates were done. In **d**, **g** data were represented as mean values ± 95% Confidence Interval. The boxes in (**b**) show the quartiles of the dataset while the whiskers extend to the entire distribution, except for distributions where outliers surpass the inter-quartile range. Statistical significance estimated by two-sided Kolmogorov–Smirnov hypothesis test with * (*p* value <0.05), ** (*p* value <0.01), *** (*p* value <0.001), **** (*p* value <0.0001). Details and exact statistical values available in the Supplementary Information.

dose-dependent effect of 375 nm excitation light dose. Higher doses led to significant increases in the number of arrested cells (a 300% increase at 60 J/cm² compared to the control), delays in the mitotic process (from an expected 20-min division to up to 50-min division times at 60 J/cm²), changes in division patterns (overall mitotic peak shifts), and a general worsening of mitotic timing (Fig. S2a–d). These observations based on manual annotation supported the results of our automated analysis, further confirming the PhotoFiTT approach. While near-UV light dose demonstrated the most pronounced effects, longer wavelengths also induced cellular stress at higher doses. Cells exposed to 475 and 630 nm excitation light began showing comparable delays to those observed with near-UV light (i.e. mitotic rounding time points above 60 min) only at the highest tested dose of 60 J/cm² while in 385 nm light excitation doses of 0.6 J/cm² was enough to cause an effect (Fig. 2d). This finding highlights a clear wavelength dependence in phototoxicity, with longer wavelengths requiring substantially higher doses to induce comparable delays, supporting observations in previously published studies[19,28]. PhotoFiTT's analysis of cell size dynamics revealed that near-UV exposure delayed the appearance of daughter cells in a dose-dependent manner (Fig. 2e). Under control conditions, mother cells (~20 μm diameter) divide into daughter cells (~15 μm diameter) within 50−75 min post-synchronisation. Near-UV light exposure induced dose-dependent delays in this process, with high doses (60 J/cm²) delaying daughter cell appearance by up to 120 min after synchronisation. Such delay in division and the persistence of cells in the rounding state over time indicates the increase in the number of cells arrested due to photodamage and explains the lost peak in the distribution of cell rounding with high doses (60 J/cm²) in Fig. 2c. Classifying mother and daughter cells according to their size lets PhotoFiTT evaluate cell division delays (Fig. 2f). When challenged with longer wavelengths (475 and 630 nm) and in agreement with our previous observations (Fig. 2d), cell division delays are also wavelength-dependant. Compared to shorter wavelengths, longer wavelengths need greater light doses to produce significant effects, such as a 20-min delay in the onset of cell division (Fig. 2f). Beyond acute effects on cell division, PhotoFiTT can also quantify post-mitotic cellular activity, providing insight into long-term impact on cellular behaviour. We observe that higher light doses and shorter wavelengths led to decreased cumulative cellular activity over a 7 h period (calculated as the cumulative activity in time) (Fig. 2g). While each of these metrics has its own benefits, tracking the 1-to-2 transition of a mother cell to two resolvable daughter cells, as done with manual annotations, provides a highly efficient and straightforward approach to assessing photodamage effects. This transition can be easily monitored by classifying cell rounding events pre- and post-division within a short time window (15−30 min). In control, synchronised populations, ~50% of cells transition from resolvable mother-to-daughter cells within the first 50 minutes post-synchronisation. At a light dose of 6 J/cm², the proportion of cells completing the mother-to-daughter transition within this time-frame is reduced to around 40% for 630 nm illumination, 30% for 475 nm, and only 10% for 405 nm (Fig. 2e, f). Testing PhotoFiTT on HeLa cells yielded similar results. Using a dose of 0.6 J/cm² did not produce discernible changes, but when challenged with higher doses of 6 and 60 J/cm², HeLa cells exhibited the same dose-dependent delay found in CHO cells. Specifically, division times increased from ~50 to 90 min, and there was a decrease in activity of up to 30% with higher excitation light doses. These findings suggest that HeLa cells might be more resilient to light damage than CHO cells (Fig. S3a–d and Supplementary Movie 4).

To evaluate PhotoFiTT's applicability across varied illumination strategies, we subsequently integrated the workflow with laser point-scanning confocal microscopy. This technique contrasts with widefield methods by scanning a focused laser beam point-by-point, thereby limiting illumination to a small volume at each instant. The duration of illumination per point, termed the dwell time, directly influences the

localised light dose and photon collection efficiency. Our objective was to determine whether distinct illumination patterns, specifically variations in dwell time and irradiance while maintaining an equivalent total light dose, elicit differential cellular responses (Fig. S4 and Supplementary Movie 5). We implemented two distinct conditions featuring the same total light dose: (1) a high-power, rapid dwell time method (fast scan), where each point is illuminated intensely for a brief period; and (2) a low-power, extended dwell time method (slow scan), where each point is illuminated less intensely but for a longer time. With this experimental design, we tested 0.6 and 6 J/cm² light doses. Cells exposed to 0.6 J/cm² presented almost no changes to their physiology with either scanning approach (Fig. S4 Fast and Slow scan at 0.6 J/cm² a−c). However, increasing the dose to 6 J/cm² yielded an increase in cell division delays from 65 to 80 min when using the slow scan, suggesting that longer excitation light exposure is more damaging than shorter exposures when normalising for total light doses (Fig. S4 Fast and Slow scan at 6 J/cm² a−c). This outcome creates an opportunity to minimise phototoxicity by fine-tuning the balance between scan speed and illumination irradiance.

Lastly, we demonstrate the applicability of the method in more realistic scenarios, accounting for multiple light exposures and the presence of excitable fluorophores. We applied the same PhotoFiTT workflow using MitoTracker dyes in a continuous fluorescence live-cell acquisition (Fig. 3 and Supplementary Movie 6). MitoTracker dyes are cell-permeant fluorescent probes that selectively accumulate in active mitochondria of live cells due to their cationic charge, which is attracted to the negative mitochondrial membrane potential[29]. The imaging workflow used RO-3306 to synchronise cell populations while adding MitoTracker Green FM and MitoTracker Red CMXRos in two different populations. MitoTracker Green FM (excitation/emission: 490/516 nm) and MitoTracker Red CMXRos (excitation/emission: 579/599 nm) were selected to evaluate wavelength-dependent phototoxic effects. These specimens were exposed to 475 nm light every 4 min during a 20-min acquisition period (150 ms per exposure event), resulting in a combined total exposure of 0.6 J/cm². This was followed by media exchange and release from the CDK1 inhibitor RO-3306, allowing cells to enter mitosis. The subsequent mitotic events were captured using only transmitted light (e.g. brightfield) to avoid further phototoxicity effects (Fig. 3a). By illustrating the timing of mitotic cell rounding in synchronised populations, Fig. 3b, c reveal a slight delay when the green fluorescent probes are excited at 475 nm wavelength. Figure 3b quantifies these delays across various MitoTracker probes, showing that the control population peaks at around 50 min. Notably, exposure to 475 nm light induced a modest delay of 15 min when exciting MitoTracker Green FM when compared to the non-exposed control. Figure 3c presents a temporal analysis of mother and daughter cell populations across different 475 nm light doses, whilst Fig. 3d depicts the cumulative cell activity over 7 h post-exposure, normalised for each replicate. The synchronised population exhibited physiological behaviour with alterations in the activity arising from using MitoTracker CMXRed and Green FM. Considering the nature of Mito-Tracker dyes and their binding to mitochondria, a small physiological change is expected[29]. Unlike conventional mitochondrial stains that wash out upon loss of membrane potential, MitoTracker dyes contain a mildly thiol-reactive chloromethyl moiety that forms covalent bonds with mitochondrial proteins, potentially affecting mitochondrial function. When assessed manually and in the absence of RO-3306, MitoTracker Green FM-stained cells also presented a modest delay in their time of division and an increased number of arrested cells (Fig. S2e, f). These results collectively underscore the sensitivity of Photo-FiTT in detecting subtle phototoxic effects during fluorescence microscopy, even when using common organelle-specific probes. Taken together, our results uncover the nuanced and continuous nature of phototoxicity. They reveal a spectrum of cellular responses that vary with both light dose and wavelength, underscoring the

## a) Imaging workflow combining PhotoFiTT and MitoTracker

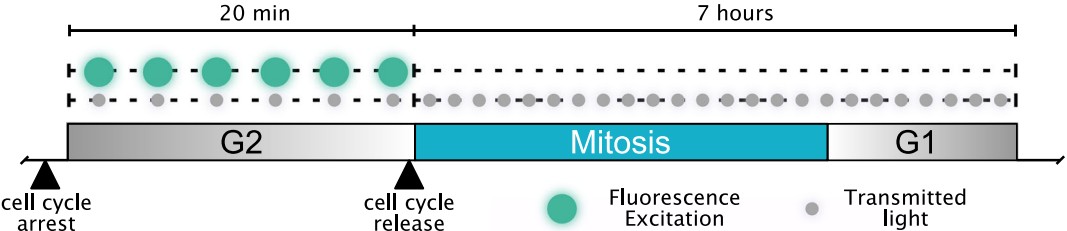

## b) MitoTracker impact on division delay

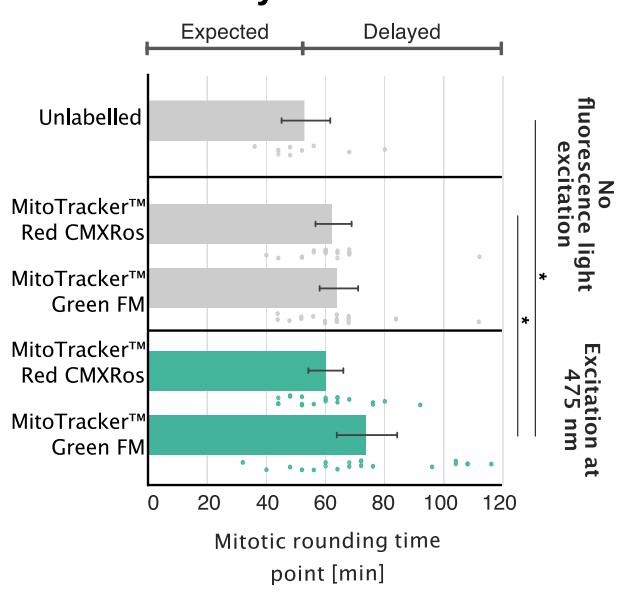

## c) Delayed Division

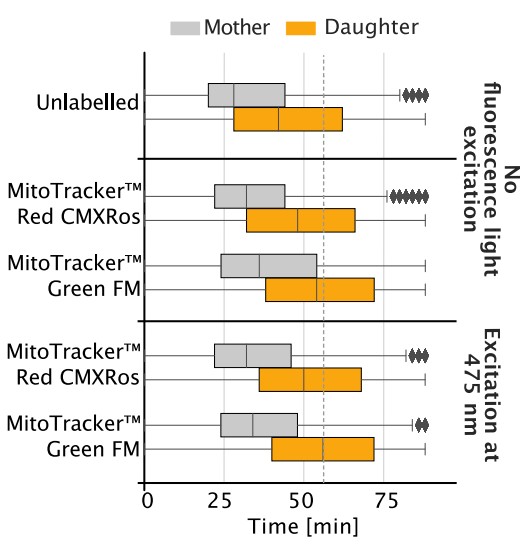

## d) MitoTracker impact on cell activity

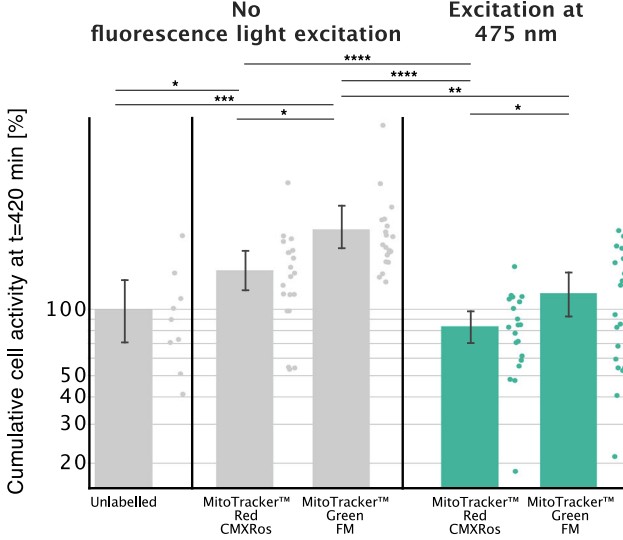

importance of considering them in experimental design (Fig. S5 and practical applications Note S3).

## Discussion

Our findings demonstrate that phototoxicity induces dose and wavelength-dependent disruptions spanning delayed mitotic progression to compromised post-division functionality, extending beyond previous observations of acute photodamage. The pronounced effects of near-UV wavelengths (385 nm) at remarkably low doses (0.6 J/cm$^2$) align with their known capacity for direct DNA interaction[30], while longer wavelengths (475 and 630 nm) required 100-fold higher irradiance (60 J/cm$^2$) to achieve comparable mitotic delays (Fig. 2d). This wavelength hierarchy persists across cell lines, with synchronised CHO populations showing 5-fold greater cell death

**Fig. 3 | Phototoxicity assessment during a fluorescence live-cell imaging acquisition.** The PhotoFiTT workflow was tested using the fluorescent probe MitoTracker. **a** Acquisition experimental design using PhotoFiTT. Live-cell imaging of MitoTracker-stained cells (green FM and CMXRed) exposed to 475 nm fluorescence excitation light (adjusted to 20 min; green) before releasing the CDK1 inhibitor drug and allowing the cells to enter mitosis. During acquisition, transmitted light (grey) was used to image brightfield. **b** Quantification of cell rounding delays across different MitoTracker probes. The control population peaks at t 50 min, representing the average cell rounding time. Exposure to 475 nm light induces a delay that becomes more apparent when MitoTracker Green FM is excited with the right light wavelength. **c** Temporal analysis of mother and daughter cell populations across 475 nm light doses. **d** Cumulative cell activity over 7 h post-exposure

normalised for each replica (logarithmic scale). The synchronised population presented physiological behaviour while both Mitotracker CMXRed and Green FM depicted changes in activity. N values correspond to each acquired field of view. Excitation at 475 nm (Unlabelled ($n = 9$), MitoTracker Red CMXRos ($n = 20$), MitoTracker Green FM ($n = 20$), MitoTracker Red CMXRos excited at 475 nm ($n = 20$) and MitoTracker Green FM excited at 475 nm ($n = 20$)). A total of two biological replicates with ten technical replicates were done In (**b**–**d**) data were represented as mean values ± 95% confidence interval. The boxes in (**c**) show the quartiles of the dataset while the whiskers extend to the entire distribution, except for distributions where outliers surpass the inter-quartile range. Statistical significance estimated by Kolmogorov–Smirnov hypothesis test with * ($p$ value <0.05), ** ($p$ value <0.01), *** ($p$ value <0.001), **** ($p$ value <0.0001).

sensitivity than unsynchronised counterparts (Fig. S1c), likely reflecting G2/M phase vulnerability to replication-associated damage. Longer wavelengths, while generally less harmful, have been described to exert significant effects on cellular physiology, including alterations in mitochondrial membrane potential, cytoskeletal dynamics and reactive oxygen species (ROS)[2,16,28]. While ROS are crucial for signalling and normal cellular functions at physiological levels, excessive amounts can cause oxidative stress, disrupting the biological processes under investigation. Namely, fluorophore excitation by light, such as those arising from fluorescent proteins or fluorescent molecular probes, boosts ROS production through their interaction with ambient oxygen, increasing the sensitivity of cells to light[12].

We implemented the statistical analysis of the results provided by PhotoFiTT, which lead to similar conclusions with statistically significant $p$ values. It should be noted that whilst significance analysis provides a valuable statistical overview, a critical interpretation of the data is essential. $p$ values are notably sensitive to empirical bias, and therefore our conclusions are drawn from a holistic assessment of the data, a principle of reproducible analysis we have previously discussed[31].

The PhotoFiTT framework quantitatively connects how illumination parameters influence distinct physiological endpoints: mitotic timing precision, daughter cell emergence kinetics, and cumulative cellular activity metrics (Fig. 2c–g). PhotoFiTT achieves this through label-free brightfield analysis enhanced by deep learning-based segmentation, avoiding fluorescence-induced confounding factors. While ROS-mediated mechanisms contribute to phototoxicity[3,32,33], our synchronisation experiments reveal phase-specific vulnerability windows that transcend simple oxidative stress models. The 300% increase in arrested cells at 60 J/cm² (Fig. S2a) underscores the need for cell cycle-aware illumination protocols.

A principal advantage lies in PhotoFiTT's capacity to establish quantitative phototoxicity thresholds, for instance, defining 6 J/cm² at 475 nm as the critical dose causing >50% mitotic delay in CHO cells (Fig. 2d). This granularity enables rational optimisation of imaging parameters rather than heuristic approaches. The framework's modular design permits extension to diverse microscopy modalities, as demonstrated through confocal dwell time comparisons where slow scanning (6 J/cm²) induced ~20% greater delays than fast scanning at equivalent doses (Fig. S4b).

Our findings demonstrate that the temporal pattern of light delivery is a critical determinant of phototoxicity, even when the total light dose is held constant. Specifically, we observed that prolonged, low-intensity illumination is more damaging to cells than brief, high-intensity exposures at equivalent total doses, challenging the conventional notion that simply lowering intensity is sufficient to minimise phototoxicity. Mechanistically, this effect is likely due to the limited capacity of cellular antioxidant and DNA repair systems, which can recover between short, intense exposures but become overwhelmed during extended or continuous illumination, resulting in greater cumulative damage.

This observation aligns with recent literature showing that phototoxicity is not dictated solely by total energy delivered, but also by

how that energy is distributed over time. Studies have demonstrated that pulsed or intermittent illumination protocols reduce cellular stress and photobleaching compared to continuous exposure[2,12,28], as recovery periods allow for more effective detoxification of reactive oxygen species and repair of sub-lethal damage. These findings advocate for the adoption of fast scanning and intermittent exposure strategies in live-cell imaging, as they better preserve cellular physiology than simply reducing light intensity.

The current implementation focuses on adherent proliferating systems, with demonstrated efficacy in CHO and HeLa models. While this covers many experimental scenarios, adaptation to non-dividing or 3D cultures remains future work. The 80% reduction in mitotic events at 60 J/cm² (Fig. S1a) highlights particular relevance for developmental studies and long-term time-lapse imaging.

PhotoFiTT's strength lies in enabling prospective, quantitative protocol optimisation rather than post hoc inference, particularly crucial for high-resolution developmental biology and cell cycle studies requiring hour-scale imaging. Future implementations could integrate with closed-loop adaptive microscopy systems, enabling dynamic exposure adjustment that reduces illumination during non-critical intervals while reserving high-dose illumination for key events like mitotic transitions or rapid organelle remodelling. This approach would balance resolution requirements with physiological fidelity, guided by PhotoFiTT's quantitative benchmarks for acceptable phototoxic load.

## Methods
### Cell lines and culture
CHO (ATCC CCL-61) and HeLa (ATCC CCL-2) cells were cultured in DMEM (Gibco) supplemented with either 42 μM gentamicin (Gibco) or 1% penicillin/streptomycin (Gibco) and 10% foetal bovine serum (FBS; Gibco). All cells were grown at 37 °C in a 5% $CO_2$ humidified incubator.

### Cell synchronisation
Cells were seeded on an eight well-chambered cover glass (Cellvis) with a total of $6 \times 10^4$ cells per well and incubated with 10 μM RO-3306 (Sigma-Aldrich) to inhibit CDK1 activity for 16–18 h. Unsynchronised cells were seeded using the same protocol and allowed to grow for the same duration.

### MitoTracker staining
Synchronised cells were incubated with media containing 10 μM RO-3306 (Sigma-Aldrich) and either 200 nM MitoTracker Green FM (Thermo Fisher, M7514) or MitoTracker Red CMXRos (Thermo Fisher, M7512) for 30 min at 37 °C.

### Excitation light dose and live-cell imaging
Experiments were performed at the Axio Observer 7 (Zeiss), equipped with a Colibri 7 LED Illumination (Zeiss), a Quad Beam Splitter 405 + 493 + 575 + 653 (Zeiss), a 20x/0.8 Plan-Apochromat objective (Zeiss) and a Prime 95B sCMOS camera (Teledyne Photometrics). The temperature and CO2 levels were maintained at 37 °C and 5% respectively, throughout the excitation light exposure and imaging periods.

Irradiance at the sample level was measured at the beginning of every experiment, using a PM100D power metre with an S170C power sensor (Thorlabs). The measurements for each wavelengths line were the following: 84.9 ± 2.8 mW for the 385 nm line, 85.2 ± 0.8 mW or the 475 nm, and 70.6 ± 0.4 mW for 630 nm (mean ± standard deviation). To convert the light energy into irradiance, we considered the field of view (FOV) size of 0.014 cm$^2$.

Cells were then exposed to the excitation light according to experimental conditions. Ten FOVs were selected per exposure condition, ensuring that exposed areas would not overlap. The exposure time was set for different fields of view in the same well according to the experimental conditions (100 ms corresponding to 0.6 J/cm$^2$, 1 s to 6 J/cm$^2$ and 10 s to 60 J/cm$^2$). Immediately after light exposure and before imaging, cells were washed two times with PBS and RO-3306 containing media was replaced by phenol-red free fluorobrite DMEM (Gibco) supplemented with 2 mM GlutaMAX (Gibco), 42 μM gentamicin (Gibco) and 10% FBS (Gibco). The media exchange was done with a custom-made 4-syringe holder designed in TinkerCAD (https://www.tinkercad.com/) and 3D printed using a Prusa MK4 3D printer (.stl file in supplementary Information). Brightfield images were acquired every 4 min for 8 h and have a pixel size of 0.55 μm × 0.55 μm.

Cells stained with Mitotracker instead underwent six sequential exposures (4-min intervals) of 475 nm light at 0.6 J/cm$^2$ total light dose, while monitoring mitochondrial dynamics (using BP 514/30 filter). Post-illumination, cells were washed thrice with prewarmed PBS before brightfield time-lapse acquisition every 5 min for 7 h.

### Cell fixation and Hoechst nuclei staining to generate fluorescent and brightfield paired images for training

Cells grown on an eight-well-chambered cover glass (Cellvis) until 60% confluency were fixed with a 4% paraformaldehyde solution (Electron Microscopy Sciences) for 10 min and washed three times with PBS. Subsequently, the samples were incubated with a 0.1 μg/ml Hoechst 33342 solution for 10 min in the dark. Cells were imaged using the same microscope setup described in live-cell imaging. Pairs of brightfield and Hoechst-stained images (the last acquired with a BP 467/24 filter in place) were captured.

### Confocal microscopy experiments

Confocal microscopy experiments were conducted using a Zeiss LSM 880 Airyscan system equipped with a 405 nm diode laser and calibrated for precise light dosing. The system's maximum output of 3.4 mW was verified using a PM100D power metre with an S170C sensor (Thorlabs). Two distinct illumination regimes were implemented to deliver equivalent total light doses of 0.6 or 6 J/cm$^2$ through compensatory power-time relationships: the high-power (HP) condition uses short 3.528 μs dwell times at 0.3 mW (8.8% max) for 0.6 J/cm$^2$ or 3.0 mW (88.2% max) for 6 J/cm$^2$, operating at 94% of the system's maximum permissible irradiance. Conversely, the low-power (LP) regime employed extended 10.584 μs dwell times (3× longer than HP) at reduced intensities of 0.1 mW (2.9% max) or 1.0 mW (29.4% max) to simulate prolonged imaging scenarios.

Key spatial parameters included a pixel dimension of 0.42 μm (0.1764 μm$^2$ area = 1.764 × 10$^{-9}$ cm$^2$), with energy per pixel calculated as the product of power (W) and dwell time (s). Both regimes maintained identical spatial sampling while achieving dose equivalence through inverse power-time compensation, enabling direct comparison of irradiance-dependent versus duration-dependent phototoxic effects. The experimental design specifically contrasted rapid high-irradiance exposure against sustained low-irradiance illumination to dissect temporal components of light-induced cellular stress.

### Live-cell viability

Cell viability was performed by adding 0.1 μM SYTOX Orange (Thermo) to previously seeded cells and taken to the microscope for live-cell imaging for 12 hours. Live-cell imaging was performed on the Nikon Eclipse Ti2 microscope equipped with a CoolLED pe800 illumination (Nikon), a 20x/0.8 Plan-Apochromat objective (Nikon) and an Orca-fusionBT camera (Hamamatsu). Ten fields of view (FOV) were randomly selected per exposure condition, and the images were acquired every 4 min using the brightfield and TRITC channels in a multichannel acquisition. The temperature and CO$_2$ levels were maintained at 37 °C and 5%.

### General image processing workflow

As depicted in Fig. 2, the image analysis to extract the measurements of interest is composed of: (1) detecting the number of cells in the field of view; (2) detecting cell rounding; (3) estimating the general changes in cell activity. These three steps enable extracting measurements for the numerical quantification of photodamage, such as cell size distribution across time.

### Estimation of the number of cells in the field of view using virtual cell nuclei staining and segmentation

The total number of cells in the field of view was computed by first, inferring the cell nuclei from brightfield images using an in-house trained Pix2Pix model[34], and then, segmenting each nucleus with a pre-trained StarDist model (StarDist-versatile) provided by the developers as '2D versatile fluo' model and trained on a subset from the Data Science Bowl Kaggle Challenge[35]. Using Zero CostDL4Mic[36], Pix2Pix was trained with paired images of fixed cells stained with Hoechst and imaged with brightfield and widefield fluorescence microscopy. The imaged data were divided into 400 and 132 fields of view for training and testing, respectively. All the input images were pre-processed as follows: a bleach correction to remove the illumination artefacts was computed by applying a large low-pass filter (a wide Gaussian filter) and subtracting it from the original image; then, the image intensity values were normalised with the min-max projection to the [0,1] range. Both pre-processed input and output images were then normalised using a percentile normalisation of 1 and 99.9%, respectively (denoted as Contrast enhancement in the ZeroCostDL4Mic Pix2Pix notebook). During Pix2Pix training, images were also reshaped to have a size of 1024 × 1024 pixels, so the images were reshaped to have a pixel size of 0.644 μm × 0.644 μm. Pix2Pix was trained from scratch with a patch size of 512 × 512 pixels, a batch size of 5 and for 2000 epochs with a learning rate of 0.001 and 1000 more epochs with a linear learning rate decay. The accuracy results for the test dataset are as follows: Structural similarity index measure (SSIM) (0.86), learned perceptual image patch similarity (LPIPS) (0.10), peak signal-to-noise ratio (PSNR) (25.17), normalised root mean squared error (NRMSE) (0.13). The output PDF reports of ZeroCostDL4Mic for both the model training and the quality control check are included in the Supplementary Information. The Pix2Pix model instance corresponding to the last epoch was chosen to process the first frame of each cell synchronisation video. The output of Pix2Pix was then segmented using the pre-trained StarDist model. The total number of individually segmented nuclei was used as the number of cells in the field of view, with an average of 126 cells per FOV.

### Cell mitosis detection with StarDist

A 2D StarDist[25] model -StarDist-CHO- was trained using Zero CostDL4Mic[36] notebooks. From all the videos acquired for the study, 105 were chosen for the creation of the ground truth. For each video, a random time-point was selected and all the rounded cells (either before or after division) were manually annotated with a unique label (i.e. instance segmentation). The data was split into 85 and 20 videos for training and testing, respectively. The model was trained on a Tesla T4 GPU for 500 epochs with patches of 525 × 512 pixels, a batch size of 20, a grid size of 2 × 2 for the first convolutional layer of the model, the Mean Squared Error (MSE) loss function and an initial learning rate of

$5 \times 10^{-3}$. Data augmentation composed of rotations, translations and mirroring was also applied. Before the training, all the images were downsampled to a pixel size of $0.865\,\mu m \times 0.865\,\mu m$ (i.e. down-sampling of a factor of 1.5709), so the size of a cell during rounding in pixels is smaller than the input size used to train StarDist. This resampling factor was chosen to ensure that the receptive field of the U-Net in StarDist covers a wide enough region to identify the cell contours characteristic of the rounded cell. The accuracy results in the test set were as follows: intersection over union (IoU) (0.530), false positive (6.05), true positive (17.05), false negative (4.6), precision (0.6712), recall (0.790), accuracy (0.555), f1 score (0.699), true detection (21.65), predicted detections (23.1), mean true score (0.712), mean matched score (0.901) and panoptic quality (0.628). The output PDF reports of ZeroCostDL4Mic for the model training and the quality control check are attached to the Supplementary Information.

## Cell activity estimation

Cell activity was calculated as the difference between consecutive pre-processed frames. First, a normalisation of the brightfield microscopy videos was applied to ensure a uniform and comparable illumination intensity across the entire video: (1) the image intensity values of each frame were normalised with the min-max projection to the [0,1] range, (2) a bleach correction to remove the illumination artefacts was computed by applying a large low-pass filter (a wide Gaussian filter) and subtracting it from the original image, (3) the image intensity values of the entire video are normalised again with the min-max projection to the [0,1]range. Second, the contrast of the video is enhanced by applying a contrast-limited adaptive-histogram equalisation (CLAHE) using the Skimage Python package with a kernel of size 25, a clip limit of 0.01 and 256 bins. Third, we applied a Gaussian filter with $\sigma = 1$ to smooth all the frames in the video and alleviate the impact of the noise in the images, which could provide wrong readings and change the activity values. Finally, for each time point t in the video, the general cell activity was computed as the difference between pairs of temporal frames as follows:

$$activity(t) = mean\left(||eIm(t) - eIm(t-1)||^2\right) \qquad (1)$$

where $eIm$ is a normalised, contrast-enhanced and smoothed image. Cumulative cell activity is calculated as the aggregation of activity ($activity(t)$) in time:

$$Cummulative\_activity(T) = \sum_{t=0}^{T} activity(t) \qquad (2)$$

## Quantitative data analysis

The percentage of cells in the field of view identified by StarDist-CHO at each time point t ($C(t)$) provides a distribution of cell rounding events across time. $C(t)$ is calculated by dividing the number of StarDist-CHO detections by the number of nuclei detected by StarDist-versatile. The peak of cell division is determined as

$$t_p : C(t_p) = \max_{\forall t \in T'}\{C(t)\} \qquad (3)$$

with $t_p$ being estimated for each video, with has a total number $T$ of time points. Cell size was monitored using the instance segmentations from StarDist-CHO (Fig. 2c, d).

The instance segmentation of StarDist-CHO allows for the estimation of the cell size (S) (i.e. the sum of all the pixels forming the cell mask), which is then used to estimate the cell diameter D by resolving the equation

$$S = \pi R^2 \qquad (4)$$

where R is the cell radius and D = 2R. Tracking the distribution of D across time we distinguish two populations: mother and daughter cells with a diameter of 20 and 15 μm, respectively (Fig. 2b, e). This is achieved by classifying all the detections with a diameter larger than 18 μm as mother cells and daughter cells otherwise (Fig. 2f).

The total n number of this analysis is the following: for 385 nm is 50 FOVs across five replicas for unsynchronised cultures, 50 FOVs across five replicas for 0 J/cm², 20 FOVs across two replicas for 0.6 J/cm², 30 FOVs across three replicas for 6 J/cm² and 30 FOVs across three replicas for 60 J/cm². For 475 nm 37 FOVs across five replicas for unsynchronised cultures, 42 FOVs across five replicas for 0 J/cm², 30 FOVs across three replicas for 0.6 J/cm², 27 FOVs across four replicas for 6 J/cm² and 14 FOVs across two replicas for 60 J/cm². For 630 nm is 44 FOVs across five replicas for unsynchronised cultures, 46 FOVs across five replicas for 0 J/cm², 12 FOVs across two replicas for 0.6 J/cm², 44 FOVs across five replicas for 6 J/cm² and 24 FOVs across three replicas for 60 J/cm². Cell activity at each time point is hardly difficult to compare due to the stochasticity of cell movement at short time windows. Therefore, we calculate the cumulative cell activity after T minutes given as

$$cumulative_{activity(T)} = \sum_{t=0}^{t=T} activity(t) \qquad (5)$$

The resulting value at $T = 420$ min (7 h) is shown in (Fig. 2g). The total n number of this analysis is the following: for 385 nm is 50 FOVs across five replicas for 0 J/cm², 20 FOVs across two replicas for 0.6 J/cm², 29 FOVs across three replicas for 6 J/cm² and 29 FOVs across three replicas for 60 J/cm². For 475 nm is 36 FOVs across four replicas for 0 J/cm², 21 FOVs across three replicas for 0.6 J/cm², 18 FOVs across two replicas for 6 J/cm² and four FOVs across one replica for 60 J/cm². For 630 nm is 26 FOVs across three replicas for 0 J/cm², seven FOVs across two replicas for 0.6 J/cm², 22 FOVs across three replicas for 6 J/cm² and nine FOVs across one replica for 60 J/cm².

## SYTOX signal quantification

To assess cellular apoptosis, we quantified the percentage of SYTOX-positive cells by dividing the number of cells expressing SYTOX by the total cell count in each field of view. The SYTOX signal was segmented using a two-step process: first, applying Otsu's thresholding algorithm, followed by a Watershed transformation to delineate individual apoptotic cells. This approach enabled an accurate approximation of apoptotic cell numbers in each frame. The image processing workflow was implemented in Fiji and automated using a custom ImageJ macro for high-throughput analysis. To determine the total cell count, we employed the method described in the previous section, 'Estimation of the number of cells in the field of view using virtual cell nuclei staining and segmentation.' A total of ten FOVs and one replica for each condition were assessed.

## Manual annotations of unsynchronised videos

A subset of unsynchronised cells was chosen to identify the delays in mitosis and cell arrest due to photodamage. Those cells entering mitosis were manually labelled by detecting the first time point in which they show the mitotic rounding until the resolvable emergence of two daughter cells. Cells that failed to produce two daughter cells by the end of the timelapse (8 h) but still attained cell rounding are considered to be cell cycle arrested. This annotation lets us distinguish between successful mitoses and cell arrest and calculate the time for cell division or arrest. A total of three FOVs per condition were annotated and tracked.

## Reporting summary

Further information on research design is available in the Nature Portfolio Reporting Summary linked to this article.

## Data availability

The data obtained in this study, as well as annotated images are available through the BioArchive [https://www.ebi.ac.uk/biostudies/bioimages/studies/S-BIAD1269] and example images to test the code are available through Zenodo [https://zenodo.org/records/12733476] both under CC BY 4.0 license. The statistical output files, as well as the source data underlying Figs. 1–3 and Supplementary Figs. 1–4 are provided as a Source Data file. Source data are provided with this paper.

## Code availability

PhotFiTT is available at GitHub [https://github.com/HenriquesLab/PhotoFiTT]. ImageJ macro to analyse SYTOX data is available at Github [https://github.com/HenriquesLab/PhotoFiTT/tree/main/IJ-macros/SYTOX]. We used ZeroCostDL4Mic to train and assess the StarDist and Pix2Pix models available on GitHub [https://github.com/HenriquesLab/ ZeroCostDL4Mic]. All source code is under an MIT License.

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

## Acknowledgements

We thank Kalina Tosheva, Buzz Baum and Caren Norden for discussions regarding experimental workflows. M.D.R., E.G.-d.-M. and R.H. acknowledge the support of the Gulbenkian Foundation (Fundação Calouste Gulbenkian), the European Research Council (ERC) under the European Union's Horizon 2020 research and innovation programme (grant agreement No. 101001332) (to R.H.) and funding from the European Union through the Horizon Europe programme (AI4LIFE project with grant agreement 101057970-AI4LIFE and RT-SuperES project with grant agreement 101099654-RT-SuperES to R.H.). Funded by the

European Union. Views and opinions expressed are, however, those of the authors only and do not necessarily reflect those of the European Union. Neither the European Union nor the granting authority can be held responsible for them. This work was also supported by a European Molecular Biology Organisation (EMBO) installation grant (EMBO-4102020-IG-4734 to R.H.), an EMBO postdoctoral fellowship (EMBO ALTF 174-2022 to E.G.-d.-M.), a Chan Zuckerberg Initiative Visual Proteomics Grant (vpi-0000000044 with https://doi.org/10.37921/743590vtudfp to R.H.) and a Chan Zuckerberg Initiative Essential Open Source Software for Science (EOSS6-0000000260). A joint collaboration between Abbelight and the Instituto Gulbenkian de Ciência kindly supports L.M. The Fundação para a Ciência e Tecnologia (FCT, Portugal) supported E.G.-d.-M. (2023.09182.CEECIND/CP2854/CT0004), and R.H. and P.M.P. through national funds to the Associate Laboratory LS4FUTURE (LA/P/0087/2020, DOI 10.54499/LA/P/0087/2020). P.M.P. and R.P. acknowledges support from Fundação para a Ciência e Tecnologia (Portugal) project grant (PTDC/BIA-MIC/2422/2020), the R&D unit Mostmicro (UIDB/04612/2020, UIDP/04612/2020). P.M.P. acknowledges support from La Caixa Junior Leader Fellowship (LCF/BQ/PI20/11760012) financed by 'la Caixa' Foundation (ID 100010434) and by the European Union's Horizon 2020 research and innovation programme under the Marie Skłodowska-Curie grant agreement No 847648, and a Maratona da Saúde award. R.P. was supported by a Fundação para a Ciência e Tecnologia (Portugal) CEEC (2023.06402.CEECIND/CP2836/CT0008, https://doi.org/10.54499/2023). This work was partially supported by PPBI—Portuguese Platform of BioImaging (PPBI-POCI-01-0145-FEDER-022122) co-funded by national funds from OE—'Orçamento de Estado' and by European funds from FEDER—'Fundo Europeu de Desenvolvimento Regional', through the Bacterial Imaging Cluster (BIC ITQB NOVA). This study was also supported by the Research Council of Finland (338537 to G.J.), the Sigrid Juselius Foundation (to G.J.), the Cancer Society of Finland (Syöpäjärjestöt; to G.J.), and the Solutions for Health strategic funding to Åbo Akademi University (to G.J.). This research was supported by the InFLAMES Flagship Programme of the Academy of Finland (decision numbers: 337530, 337531, 357910 and 357911). This work was partially supported by PPBI—Portuguese Platform of BioImaging (PPBI-POCI-01-0145-FEDER-022122), co-funded by national funds from OE - 'Orçamento de Estado' and by European funds from FEDER—'Fundo Europeu de Desenvolvimento Regional'.

## Author contributions

R.H. and P.M.P. conceived the project. M.D.R., E.G.-d.-M. P.M.P. and R.H. designed the experimental and analytical pipelines. M.D.R. and L.M. performed the live imaging experiments. M.D.R., E.G.-d.-M. and R.P. acquired the brightfield-fluorescent paired data. M.D.R., E.G.-d.-M. and L.M. annotated and curated the data. E.G.-d.-M. developed the analytical pipeline. M.D.R. wrote the Fiji macros. E.G.-d.-M. and M.D.R. analysed the data with inputs from G.J., P.M.P. and R.H. M.D.R. and E.G.-d.-M. prepared and tested the GitHub repository and wrote the paper with input from G.J., P.M.P. and R.H. All authors reviewed and refined the manuscript.

## Competing interests

The authors declare no competing interests.
