## [Transparent Peer Review file · Nature Communications]

PhotoFiTT: A Quantitative Framework for Assessing Phototoxicity in Live-Cell Microscopy Experiments

Corresponding Author: Professor Ricardo Henriques

Version 0:

Reviewer comments:

Reviewer #1

(Remarks to the Author)

Del Rosario and colleagues present a framework that integrates a standardized experimental protocol with advanced imaging analysis to quantify light-induced cellular stress. This work is particularly timely, as optical microscopy has become one of the most important tools for observing biomolecular processes in living cells. However, it is crucial to understand when the processes under study are influenced or altered by the observation itself—specifically by the light used to capture the images. For this reason, I am supportive of its publication in Nature Communications, as the journal's broad readership aligns well with the relevance and significance of this report.

That said, before publication, I believe there are a few points that need to be clarified or described in greater detail:

- The framework is presented as a method to quantify phototoxicity in fluorescence microscopy, but this aspect is not clearly emphasized in the text. If I understand correctly, the three main measurements are performed on brightfield images, while fluorescence measurements are only used for generating the training set for virtual staining. Although the method's applicability to non-fluorescent cells is an advantage, it may cause some confusion. Specifically, is the "Acquisition" phase (Fig. 2a) conducted on fluorescently labeled cells? This does not necessarily imply that fluorescence images are used in the analysis itself, and clarifying this distinction would help improve understanding of the method.
- I would expect the degree of cellular stress to differ in fluorescently labeled cells, as fluorophores absorb light, reducing cell transparency compared to non-labeled cells. The authors seem to acknowledge this, as they mention light absorption in the context of the CDK1 inhibitor. However, I recommend further clarification of how labeling affects light absorption and, consequently, cellular stress, and how this factor may influence experimental outcomes.
- The cellular activity measurement is presented as a method to quantify sub-cellular changes. However, I believe this claim may be overstated. In essence, the measurement seems to reflect changes in cell shape rather than directly assessing alterations in the behavior of sub-cellular components, such as mitochondria or vesicles. I suggest refining this claim to more accurately describe what the measurement captures.
- Can you better explain why measurements need to be performed on both synchronized and unsynchronized cells? Is this an integral part of the protocol, or are these just two examples? It would be helpful to clarify whether this dual approach is essential for the method or simply demonstrates its flexibility.

Other minor issues include:

- Could you indicate the size of the field of view (FOV) used to convert power into intensity?
- There are small typos: "uu" instead of "µm" appears in multiple instances on page 6.

(Remarks on code availability)

Reviewer #2

(Remarks to the Author)

In this manuscript, the authors present an application of pre-existing deep-learning based tools to assess the photodamaging effects of light illumination on live cells. Using non-damaging bright field illumination, the authors record the behaviour of illuminated cells over many hours, using software tools to segment and analyse cells. Three measurements are used to assess photodamage: delays in mitosis, identification of cell cycle arrest, and levels of cellular activity. This is applied to Chinese Hamster Ovary (CHO) cells, exposing them to UV, blue or red light. For UV illumination, delays in mitotic

timing were observed with doses as low as 0.6 J / cm^2 . For red illumination, only 60 J / cm^2 was sufficient to cause noticeable damage.

The manuscript is clear and well-written. The supplementary videos provide a nice overview of the work, but video 2 is too fast — it is impossible to follow what is being pasted into the terminal, etc.. Figures 1 and 2 provide a nice description of the assessment pipeline, but it would be nice to include images of 'bad' cells as well as 'happy' cells. The authors should be commended for making their pipeline available as well-documented open-source code. I have, for reasons of time, been unable to install and operate the code on my own computer, but given the authors' track record in developing software I am confident that the software would operate correctly if I had. However, if I have understood correctly, applying to the pipeline to cells different to the ones used by the authors (CHO), then the neural networks employed (developed by other researchers) must be retrained. This significantly restricts the utility of the tool.

The manuscript consists of two main parts: 1) the development of a software pipeline for analysing photodamage, 2) the application of this approach to studying photodamage in live cells. While I think the first part may have general utility for many researchers, the second part is not very thorough. I certainly would not draw any general conclusions from these results. This is important, as the presented results contradict many imaging studies where similar irradiances have been used without ill effects. Indeed, Wäldchen et al. (2015) have shown that cells can withstand even many kJ / cm^2 of red illumination, while the authors here suggest that just 60 J / cm^2 is enough to damage them.

One possible reason for this discrepancy is that the methods section states that cells were cultured in DMEM, and this was only switched for phenol-red-free Fluorobrite DMEM after photodamaging illumination. This is very different from the usual scenario in which cells would be illuminated in imaging medium (such as Fluorobrite DMEM). Phenol red absorbs strongly at all wavelengths used and has previously been reported to significantly increase photodamage from imaging illumination. Another possibility is that Wäldchen et al. used a range of U2OS, COS-7 and HeLa cells, while here the authors used CHO cells. If a general claim about photodamage is to be made, these experiments must be repeated with a different cell line in order to ensure that the photodamage is not a CHO-specific effect.

The authors report illumination powers, but it is not clear how these were measured. If a standard power meter sensor was used then this is likely to be an underreport, as these power meters are not sensitive to the high-angle rays present in microscopic illumination. Instead, a specialist power meter, such as the Thorlabs S170C, should be used (and if this was used this should be clearly stated in the text). Furthermore, it is not clear how these powers were converted to irradiances. Different powers were used for different wavelengths, but this apparently led to the same irradiance. This can only be true if the field of illumination varies by wavelength and it is not clear that this should be the case. Indeed, it appears (by backcalculation) that the authors calculated the irradiance by dividing the power by the full field size of the objective ($\pi * (12.5 \text{ mm} / 20)^2$). This assumes that there is no illumination outside of the manufacturer's claimed field of view, which is not true (as there is no real field stop in the objective). Instead, an external field stop of known size should be inserted into the illumination path.

Perhaps most critically, the illumination timing used in the mock illumination is very different from that used in real timelapse experiments. In this work, the illumination is delivered in a single continuous dose at the beginning of the experiment. In a typical live-cell imaging experiment, the illumination would instead be delivered in little chunks throughout. Delivering a large dose up-front will create a much larger number of reactive oxygen species and other undesirable photoproducts that may overwhelm a cell's protective mechanisms in a way that a more distributed photodose would not. I strongly suggest that the authors perform experiments using their analysis framework where cells have been illuminated in a more realistic manner. Only then, in combination with further experiments using multiple cells lines and different media, will the results allow general conclusions to be drawn.

Given the currently narrow applicability of the developed pipeline (i.e. only applicable to CHO cells), and the aforementioned issues with the experiments, I feel unable to recommend this manuscript for publication in Nature Communications. However, if the pipeline can be made more general and the experiments made more robust and generalisable, then I see that this could become a valuable tool for the community. Some minor comments are detailed below:

== Minor comments ==

* Given that RO-3306 is known to cause apoptosis after long incubation times, have the authors considered using a different means of cell cycle synchronisation, such as tapping a plate to dislodge only mitotic, rounded cells? This would provide a smaller number of cells for further analysis, but it seems that they are not analysing an entire cell culture plate anyway. Importantly, this would completely rule out the possibility that the effects observed were related to RO-3306.

* Can the authors comment on how the use of point-scanning illumination, such as that employed in confocal microscopes, might alter the conclusions drawn from the epi-illumination used in this work?

* No line numbers are included in the manuscript file. While not strictly necessary, these would make it easier for reviewers to refer directly to specific parts of the text.

* What level of mitotic delay is enough? Is a 25 minute delay biologically significant for CHO cells? What about other cell types?

* Related to the last point, there is no indication of statistical or biological significance between the different conditions tested, using any of the metrics employed.

* The authors' results suggest that live-cell single molecule localisation microscopy (SMLM) is impossible without perturbation of the cell's health, given that a 60 J / cm^2 dose is sufficient at 630 nm, while irradiances of the order of hundreds of W / cm^2 are used. How can this view be married with the extensive literature showing successful live-cell SMLM? Have these imaged cells all been irreparably damaged without anyone noticing?

(Remarks on code availability)

I have successfully accessed the code repository on GitHub and reviewed the documentation and notebooks, but have not had time to install and run it myself. The code appears to be well-written and clear, as does the documentation.

Reviewer #3

(Remarks to the Author)

The work introduces PhotoFiTT, a standardized workflow for quantitatively evaluating the phototoxic effects on adherent and dividing tissue culture cells. The workflow utilizes machine learning tools to enhance cell segmentation from brightfield transmission microscopy time series and extract three different morphological parameters and their changes over time, which indicate cell fitness, particularly the ability of timely cell division. The authors demonstrate the application of their approach on CHO cells that were exposed to three different light doses at three different wavelengths, showing the expected correlation in the quantitative evaluation parameters.

Overall, the work is very well presented, and the tool is potentially useful to be adopted by other labs to optimise experimental conditions for live cell microscopy studies.

However, I feel it currently falls short of demonstrating its validity in real-world scenarios, as I will detail below. Thus, before recommending this work for publication, I would like to see the following issues and queries addressed.

Major issues:

1. The authors evaluate the phototoxic effects of 3 different light doses that deviate by a factor of 10, at three different wavelengths to validate their approach. It seems that the effects of UV-light and higher light doses on the cell population are obvious even without a sophisticated quantitative evaluation. Can the approach reliably detect and quantify more subtle effects? Are there limitations to doing this?
2. The PhotoFiTT assay is per se label-free, which is beneficial as it should not contribute to phototoxicity. However, in most realistic live cell imaging scenarios, cells are fluorescently labeled and observed at high resolution over a certain observation period with a certain temporal resolution. Importantly, bleaching of fluorophores itself generates reactive oxygen species that contribute to phototoxicity. Therefore, it is essential to determine the phototoxic effects in the specific experimental conditions where labelling is applied and observed with the same objective. This allows for the comparison of phototoxic effects with the baseline or with different illumination regimes. Eventually, this will inform the researcher on the data amount, the level of resolution, and SNR quality needs can be afforded without inducing unwanted biological side effects that otherwise might affect the interpretation of the experimental outcome. Showcasing the application of PhotoFiTT to one such realistic example is needed to be more convincing about its broader value.
3. Since the workflow essentially assesses effects on cell division rates, why not simply count the increase in cell number density in the 10 irradiated areas vs 10 non-irradiated areas before and, say, 24 hours after irradiation? Also, what is the benefit of assessing three different division-linked morphological parameters? Are there specific situations where this redundancy is beneficial?
4. An obvious shortcoming of the PhotoFiTT assay (as is also mentioned by the authors), is its restriction to dividing cells, as all three metrics effectively measure mitotic or postmitotic behaviour. Could an activity measurement alone not also be adapted to work for non-dividing, differentiated cells like neurons?
5. The authors mention the transferability of the assay to other widefield (?) imaging systems. Does this also apply to confocal systems? If so, this should be explicitly mentioned and ideally be demonstrated. For example, a low-hanging fruit might be to quantitatively assess the difference between delivering the same light dose by a widefield illumination (low power density, longer dwell time) or by the focused laser beam of a confocal point scanning system (high power density, short dwell time), which has been a long-standing point of discussion in the field.

Minor issues:

The text mentions that cell nuclei are segmented, while the figures give the impression that entire cells are segmented. This needs clarification.

Which objective was used for the irradiation? Is it the same as for imaging?

How big were the FOVs accordingly? How many cells are typically irradiated and assessed in a single experiment?

The NA of the objective should affect the energy distribution within cells, with a high NA compressing more energy into a smaller focus z-range? Has this been accounted for?

What is the energy (J/cm²) for the transmission observation compared to the excitation light irradiation? Has it been controlled that this does not influence the “baseline” physiology?

Did the authors control for potential cytotoxic effects of RO-3306 independent of light exposure?

(This may only apply to my downloaded file) Video S1 stops prematurely at 01:44 of the 2:19 total runtime.

Typo on p6, 2nd column: “...the images were reshaped to have a pixel size of 0.644 μ x 0.644 μ m.”

(Remarks on code availability)

Reviewer #4

(Remarks to the Author)

In this paper, the authors present a label-free and quantitative approach to evaluate phototoxicity for live-cell imaging experiments. The aim is to enhance the reliability and reproducibility of such live-cell imaging studies. More approaches to quantify phototoxicity are welcome - especially updated ones that exploit recent technical advances. The strengths of the presented study lie chiefly in using a label-free (brightfield) approach, a large amount of data that can be (semi-) automatically processed for quantification, and the use of trainable open-source software. This has the potential to make it widely applicable (within the area of live-cell imaging, and depending on the uptake of its community). The authors use three parameters: Duration of mitosis (the ideal ‘canary in the gold mine’, Cole 2014), cell size and ‘cumulative activity’. Manuals, videos and curated code are a very useful addition to aid replication and adaptation by others.

Below are some suggestions to improve the manuscript, and some issues that need addressing.

Major comments:

1) As presented here, we only see this method applied to one type of cells and one type of microscopy. Perhaps this could be expanded. For example, does the presented approach work for any optical contrast method (phase contrast (as used) as well as differential interference contrast (DIC))? Was this method tested on e.g. motile cell lines? This expansion is optional, but would prove wider applicability. Alternatively, conclusions and claims should be modified. “Addressing this gap, we introduce PhotoFIT, a quantitative imaging-based framework designed to assess phototoxicity effects on cellular behaviour in live-cell microscopy experiments.”
– for adherent cell lines in traditional slide-based setups.

2) It would be useful to show how well the algorithm fares compared to a ground truth. The ground truth may likely require manually determined mitotic durations and segmentations. However, without any comparison, the question of accuracy remains unanswered.

3) The authors' results nicely illustrate aspects of phototoxicity that have been previously published. The way it is currently written, it is not always clear whether the novelty of this submission lies in the methods used or in (largely unreferenced) observations. It would be good to more clearly promote the first aspect, while ensuring that stated observations align well with previously published studies.

A few omitted references are used below as examples. While not exhaustive, and not all recent, these are important papers and good starting points for the reader/authors.

Mitotic delay / using time spent in mitosis as measure:

Cole R. 2014. Live-cell imaging. *Cell Adhes Migr* 8: 452–9.

Gorgidze LA, Oshemkova SA, Vorobjev IA. 1998. Blue light inhibits mitosis in tissue culture cells. *Biosci Rep* 18: 215–24.

Mora-Bermudez F, Ellenberg J. 2007. Measuring structural dynamics of chromosomes in living cells by fluorescence microscopy. *Methods* 41: 158–67.

Wavelength dependence of phototoxicity:

Douthwright S, Sluder G. 2017. Live cell imaging: assessing the phototoxicity of 488nm and 546nm light and methods to alleviate it. *J Cell Physiol* 232: 2461–8.

Schneckenburger H, Weber P, Wagner M, Schickinger S, et al. 2012. Light exposure and cell viability in fluorescence microscopy. *J Microsc* 245: 311–8.

Image processing to classify phototoxicity from transmitted light images:

Richmond D, Jost AP-T, Lambert T, Waters J, et al. 2017. DeadNet: identifying phototoxicity from label-free microscopy images of cells using deep ConvNets. *arXiv* 170106109.

Grah JS, Harrington JA, Koh SB, Pike JA, Schreiner A, Burger M, Schönlieb CB, Reichelt S. Mathematical imaging methods for mitosis analysis in live-cell phase contrast microscopy. *Methods*. 2017 Feb 15;115:91-99. PMID: 28189773.

Cell outline changes:

Knoll SG, Ahmed WW, Saif TA. 2015. Contractile dynamics change before morphological cues during fluorescence

illumination. Sci Rep 5: 18513.

Re “critical need for standardised and generalisable methods able to quantitatively link cell damage to high-intensity light exposure across different fluorescence microscopy techniques.” Important additional publications:

Kiepas A, Voorand E, Mubaid F, Siegel PM, Brown CM. Optimizing live-cell fluorescence imaging conditions to minimize phototoxicity. J Cell Sci. 2020 Feb 21;133(4):jcs242834. PMID: 31988150.

Alghamdi RA, Exposito-Rodriguez M, Mullineaux PM, Brooke GN, Laissue PP. Assessing Phototoxicity in a Mammalian Cell Line: How Low Levels of Blue Light Affect Motility in PC3 Cells. Front Cell Dev Biol. 2021 Dec 17;9:738786. PMID: 34977004.

4) An important advantage of the proposed method may be its sensitivity. However, the authors are not clear on their method's sensitivity, to the point of contradicting themselves:

p. 4, middle left:

“A key strength of PhotoFiTT is its ability to reveal the non-binary nature of phototoxicity, demonstrating a continuum of effects that vary with light dose and wavelength in a label free manner.”

Versus p.4 bottom left:

“It is important to acknowledge limitations of PhotoFiTT. While highly sensitive to major cellular perturbations, it may not detect subtle light-induced events.”

It is important for this publication to have an unambiguous statement regarding its method's sensitivity (or lack thereof) and evidence it.

Cellular morphology alone is not a clear indicator of safe illumination levels. The physiology of living specimens can be altered already at much lower illumination levels. This subtle phototoxicity, affecting healthy-looking specimens, can change the kinetics of cellular and developmental processes, leading to data misinterpretation. Dynamic processes have been shown to be sensitive indicators of phototoxicity (Kiepas 2020, Alghamdi 2021, Icha 2017) compared to morphology, and can indicate positive (hormetic) effects.

With this in mind, it might also be useful to revisit the statement “...integration of PhotoFiTT principles into real-time analysis pipelines to apply corrective measures to avoid cell death.”, as this goes back to the binary definition of phototoxicity.

5) On p.1, the authors state: “Importantly, tolerance to photodamage varies across specimens and is influenced by damage severity, complicating the development of replicable imaging protocols (12, 13)”.

Later, they write that “a critical need remains for standardised and generalisable methods able to quantitatively link cell damage to high-intensity light exposure across different fluorescence microscopy techniques.”

It might be worth expanding on this. A method to assess phototoxicity can only be standardised and generalised to a limited degree, since phototoxic effects always depend on the sample (and many other factors such as media, type and localisation of fluorescent proteins used etc.). See Waldchen et al. 2015 with ~6-fold difference of phototolerance between HeLa cells and U2OS (or COS-7) cells.

For the same reason, providing a hard limit (“For imaging domains where illuminations are well above the 60 J/cm² threshold studied here, researchers should carefully consider and mitigate photodamage.”) is dangerously reductive and misleading. In phototoxicity, there is no 'one size fits all'. For a large variety of samples, photodamage may affect samples long before this threshold.

Minor comments:

The instructions are not sufficient in parts - e.g. measuring the power at the sample would benefit from more detail in the methods section and the video.

What would also be very helpful: Guidelines on how many images need to be taken to ensure sufficient spatial and temporal resolution (at least in the case of the cell line used here), and how many cells analysed to reach robust conclusion(s).

“Cumulative cellular activity” is not well defined. It needs to be clearly defined in the main text. How is it distinct from cell diameter dynamics, and what does it signify? Also, why ‘cumulative’? This is used earlier for phototoxic effects, but for cellular activity seems to be with respect to ‘change of area over time’. Also, is cell activity directional (i.e. only shrinkage taken into account)?

Also, be clear on the definition and units of irradiation vs irradiance and intensity (e.g. “Table S1: Irradiation range (W/cm²)”).

p. 1 bottom left: “Traditional methods for assessing phototoxicity include viability assays and morphological observations.” Requires some references.

Fig. 1 B:

Cells here look elliptical, but diameter calculation is for perfectly round cells. If an ellipse is fitted, indicate whether long or short axis is chosen (or result discarded).

“Expose cells to a light irradiation event replicating the illumination pattern of the imaging

experiment to be analysed for phototoxicity;" - phototoxicity

"...purposely-trained": correct the first word.

"Video S1-2 provide in-detail tutorials to reproduce these procedures." Change to: "Videos S1 and S2 provide..."

p. 2 top left:

"low-illumination brightfield microscopy" – meaning low-intensity?

Also, what is subcellular about Fig. 1 C? The outline demarcates the cellular border.

p 2. top right:

"Notably, synchronised populations exhibited higher sensitivity to near-UV-induced damage compared to their unsynchronised counterparts." Read e.g. Icha et al. 2017 for references on this.

"the persistence of mother cells upon time" - unclear

"Upon challenged" - unclear

"with longer wavelengths (475, and 630nm)" - change to: 475nm and 630nm

"cell division delays are also wavelength-dependant, requiring higher light doses to induce noticeable effects". Not clearly formulated - why comparative? Do the authors mean 'higher light doses at longer wavelengths'? Also, 'noticeable effects' - meaning delays? Clarify.

Suppl. mat. pdf, p. 2:

"Recomentation" - Recommendation

"For other cell lines: For other cell types or experimental conditions, manually annotate a representative image set and train a new StarDist model following the protocol outlined in Methods Section D." Which is Methods Section D? This one?

Unclear.

video 710560_sh8m2f: There's 30 seconds of a frozen frame at the end of the video.

video 710566_sh8sb8: Too fast to follow - can it be slowed down please.

video 710567_sh8sbm: nice illustration of effects; however, blue shading at bottom overpowers the plot line.

(Remarks on code availability)

Version 1:

Reviewer comments:

Reviewer #2

(Remarks to the Author)

I commend the authors for their efforts in revising the manuscript, which have addressed many of the points raised by myself and the other reviewers. The addition of the confocal scanning and HeLa experiments is welcome, and it is nice that the authors could retrain the neural networks used to assess HeLa cell data without much effort. While I still have concerns that the results provided by PhotoFITT seem at odds with those previously reported in the literature, the revised text and experiments make it clearer that this is a tool that can be used to optimise a particular experiment, rather than one that can be used to study phototoxicity in general. Indeed, I remain unconvinced that mitotic delay is a relevant measure of damage for many samples, but that is up to the user to decide. As such, I would be happy to see the manuscript published in Nature Communications.

(Remarks on code availability)

The code is clearly written and well-documented.

Reviewer #3

(Remarks to the Author)

The authors made an excellent effort to respond to my and the other referees' criticisms and implement their suggestions, thus significantly improving the quality of the manuscript. However, I still have a few minor remaining questions and comments that I would like to see addressed before ultimately recommending the work for publication.

(1) Statistics: The figure legends should indicate whether error bars in diagrams show standard deviations or standard errors.

Related to this, I don't fully buy the argument given in response to Reviewer #2's request for significance tests, that those would not add any insight. An alternative might be to indicate the level of significance (by number of star signs) and briefly

discuss the limitations for interpretation in the methods section. It also begs the question of why the sample sizes were not appropriately chosen in the first place to achieve valid significance.

(2) Line 207ff: "These specimens were imaged for 20 minutes using a constant light exposure every 4 minutes for a total acquisition of 20 minutes with a combined exposure of 0.6 J/cm²." This sentence reads somewhat redundantly. What was the actual effective time of exposure? Would it matter if the effective exposure time is applied continuously for a shorter time or with intervals over a longer period?

(3) Table 1 and more generally: From assaying two different cell lines, CHO and HeLa cells, it would be useful to present an example calculation on how to derive a starting point or recommendation for users who work with those cell lines. For example, for wide-field live-cell imaging of a green-fluorescent target in HeLa cells, setting the illumination to a 5 W/cm² power density would reach a "safe dose" of 5 J/cm² with a total acquisition time of 1 s. With 10 ms exposure time (provided this yields sufficient SNR), this would allow for distributing the dose over 100 exposures, which can be used to either image 100 time points for a 2D time series, or correspondingly fewer time points for a 3D time series. Decreasing the power density by 10-fold, and perhaps using DL denoising to overcome SNR penalty, would allow 1000x 10-ms-exposures, while adding a colour channel would reduce the number of afforded time points again, depending on the wavelength (not sure if my back of the envelope calculation is correct, but something along those lines).

(4) There are still many typos that will likely be amended in the editorial process. However, one that might be prone to being overlooked is "Hoeschst", which should read "Hoechst".

(Remarks on code availability)

Reviewer #4

(Remarks to the Author)

There are two major issues that need to be resolved:

1) L 235: low doses (0.6 J/cm²) align with their known capacity for direct DNA interaction³⁰, while longer wavelengths (475 nm, 630 nm) required 100-fold higher irradiance (60 J/cm²)

> As requested for the first revision, please be clear on the definition and units of irradiation vs irradiance. Irradiance is a measure of the power of electromagnetic radiation (such as light) that is received over a given area. The SI unit for irradiance is watts per square meter (W/m²) - of course, variations thereof can be used. How are the authors using the term irradiation? As exposure / fluence / light dose (J/cm²)? A useful table can be found in ref. 5. Note that in the context of microscopy, the term intensity is often (albeit incorrectly) used interchangeably with irradiance.

2) PhotoFiTT analyses three cellular features: Rounding, size (large mother cells, smaller daughter cells) and activity (cellular outlines).

Rounding is always called 'mitotic rounding' - but cellular rounding can be a stress response and due to cellular swelling (osmotic imbalance) and loss of homeostasis. For it to be called 'mitotic rounding', the rounding has to be one cell separating into two that is being assessed.

Indeed, in line 160, the authors write: While each of these metrics has its own benefits, tracking the 1-to-2 transition of a mother cell to two resolvable daughter cells provides a highly efficient and straightforward approach to assessing photodamage effects.

But it is unclear if PhotoFiTT uses tracking / pedigree analysis as a default analysis, or if this is an additional analysis used in this study. At what point does the study switch from generic rounding/cell size metrics to the 1-to-2 transition analysis?

This needs to be made very clear for the reader.

Also, line 62 states that 'Cellular activity reflects general dynamic changes in cellular shape across time.' How is this differentiated from mitotic rounding?

A third point regards intensity vs. exposure time.

In line 184, the authors state: longer excitation light exposure is more damaging than shorter exposures when normalising for total light doses (Figure S4 Fast and Slow scan at 6 J/cm² a-c). This outcome creates an opportunity to minimise phototoxicity by fine-tuning the balance between scan speed and illumination intensity.

and in line 456: ...dose equivalence through inverse power-time compensation, enabling direct comparison of intensity-dependent versus duration-dependent phototoxic effects. The experimental design specifically contrasted rapid high-intensity exposure against sustained low-intensity illumination to dissect temporal components of light-induced cellular stress.

This is a very interesting finding that in my view would benefit from being discussed more in-depth. Embedding this result in the wider literature (studies reducing intensity vs exposure time, and/or studies finding that only the total light dose determines the degree of phototoxicity, independent of how light is delivered) addresses important practical considerations for a wide readership.

L 31: comparable effects.

> comparably deleterious effects.

L 42: photobleaching has been used as a proxy reporter of phototoxicity¹²

> far more publications than 12 have used this. As 2 and 5 are reviews, adding these should be enough.

L 44: Importantly, tolerance to photodamage varies across specimens and is influenced by damage severity
What do the authors mean? This is circular argumentation.

L 46: Previous studies have offered valuable quantitative assessments of phototoxicity^{3,10,14,16,17}

> 12,25,28 are also valuable quantitative assessments. They should also be referenced.

Fig.1 caption: biological imaging assay

> what do the authors mean by 'biological' imaging assay? In the context of phototoxicity, it will have to be biological.

L 66: It detects deviations in these patterns caused by light exposure, enabling the quantification of phototoxic effects in fluorescence microscopy.

> This is only the case where a dependence on light dose (or irradiance) has been demonstrated. The patterns can also be caused by other factors (temperature, media, cumulative damage etc).

Fig. 2 caption, line 104: t 50 minutes

> remove t

L 276: The PhotFiTT computational framework

> PhotoFiTT

L 168: challenged with higher doses of 6 and 60 J/cm², HeLa cells exhibited the same dose dependent delay

> dose-dependent delay

L 438: Cell fixation and Hoescht nuclei

> Hoechst, not Hoescht. Replace throughout.

L 510: attached to the supplementary material.

> 'attached to'?

L 522: alleviate the impact of the noise in the images

> unclear

L 527: where *elm* is a normalised, contrast-enhanced and smoothed image. Cumulative cell activity is calculated as the aggregation of activity (*activity(t)*) in time:

> italicise *elm* and *activity(t)*

L 530: Cumulative

> Cumulative

L 539: with *t_p* being estimated for each video, *V*.

> *V* is not in the equation.

L 573: The image processing workflow was implemented in Fiji and automated using a custom ImageJ macro for high-throughput analysis.

> Indicate the availability of the macro.

(Remarks on code availability)

We are profoundly grateful for your thorough evaluation of our manuscript. We have meticulously addressed each point raised, implementing comprehensive revisions that strengthen our findings and broaden their applicability.

The revised manuscript now incorporates several new experimental expansions. We have extended our cellular model validation by including HeLa cells alongside the original CHO line, providing critical cross-validation that demonstrates the robustness of our PhotoFiTT framework across diverse mammalian systems. This comparison revealed intriguing differential photosensitivity profiles, with HeLa cells exhibiting greater resilience to light-induced damage than CHO cells at equivalent doses yet maintaining consistent dose-dependent response patterns. We have also thoroughly examined the phototoxic contribution of fluorescent probes through targeted MitoTracker experiments. These investigations specifically quantify how fluorophore excitation amplifies cellular photodamage, demonstrating that even at modest illumination doses (0.6 J/cm²), MitoTracker Green FM produces measurable 15-minute delays in mitotic timing when excited at 475 nm. This observation aids in using our assay to inform optimal probe selection for imaging protocols. Furthermore, we have conducted confocal microscopy experiments comparing varied illumination regimes at equivalent total doses. By contrasting fast scanning (3.528 μs/pixel) with slow scanning (10.584 μs/pixel) at identical energy inputs (6 J/cm²), we revealed that dwell time influences photodamage manifestation. These findings provide practical guidance for parameter optimisation in point-scanning microscopy applications.

We sincerely appreciate the opportunity to enhance our manuscript and believe the resulting work represents a substantial advancement in understanding and mitigating phototoxicity in quantitative microscopy.

Please find below our point-by-point reply to reviewers:

Reviewer #1

Reviewer #1: Del Rosario and colleagues present a framework that integrates a standardized experimental protocol with advanced imaging analysis to quantify light-induced cellular stress. This work is particularly timely, as optical microscopy has become one of the most important tools for observing biomolecular processes in living cells. However, it is crucial to understand when the processes under study are influenced or altered by the observation itself—specifically by the light used to capture the images. For this reason, I am supportive of its publication in Nature Communications, as the journal's broad readership aligns well with the relevance and significance of this report.

That said, before publication, I believe there are a few points that need to be clarified or described in greater detail:

Our reply: We sincerely thank the reviewer for recognising our work and their dedication to providing valuable feedback. As follows, we address directly each of the comments made.

- The framework is presented as a method to quantify phototoxicity in fluorescence microscopy, but this aspect is not clearly emphasized in the text. If I understand correctly, the three main measurements are performed on brightfield images, while fluorescence measurements are only used for generating the training set for virtual staining. Although the method's applicability to non-fluorescent cells is an advantage, it may cause some confusion. Specifically, is the "Acquisition" phase (Fig. 2a) conducted on fluorescently labeled cells? This does not necessarily imply that fluorescence images are used in the analysis itself, and clarifying this distinction would help improve understanding of the method.

Our reply: PhotoFiTT aims to assess the phototoxicity caused by fluorescence light without relying on fluorescent markers or additional illumination to avoid measurement biases. Thus, the "Acquisition" step in Figure 2a only involves transmitted light microscopy. The source of damage, such as fluorescent excitation light illumination, is meant to be applied before in what we call the "Photodamage irradiation" step. The description in the text has been changed to "This implementation (Figure 2a), followed these key steps: 1) Synchronise cells using the CDK-1 inhibitor RO-3306, which arrests cells in the G2/M interface, or study unsynchronised cells that can be tracked from mitotic rounding until division; 2) Expose cells to a light irradiation event replicating the illumination pattern of the imaging experiment to be analysed for phototoxicity; 3) Identify cells undergoing mitotic rounding and division (Figure 2b) using a Stardist deep learning model purposely-trained for brightfield round cell detection; 4) Analyse three key parameters: mitotic timing, cell diameter dynamics, and cellular activity". Fig. 2a has also been updated and the caption now reads as "Synchronised cells are washed prior to 6-hour time-lapse imaging, including transmitted light and fluorescence markers for model training. Mitotic rounding is identified using trained models to quantify cell activity and division dynamics. Detailed workflow descriptions are in Notes S1 and S2".

Reviewer #1: I would expect the degree of cellular stress to differ in fluorescently labeled cells, as fluorophores absorb light, reducing cell transparency compared to non-labeled cells. The authors seem to acknowledge this, as they mention light absorption in the context of the CDK1 inhibitor. However, I recommend further clarification of how labeling affects light absorption and, consequently, cellular stress, and how this factor may influence experimental outcomes.

Our reply: Following also the recommendation of Reviewer 3, we now include new experimental results (new Fig. 3, Figure S2 e,f, Video S6) assessing the effects of fluorescence tags (MitoTracker), which show increased cellular damage, as

highlighted by the reviewer. Additionally, we have expanded the discussion in the manuscript by adding the following information: “While ROS are crucial for signalling and normal cellular functions at physiological levels, excessive amounts can cause oxidative stress, disrupting the biological processes under investigation. Namely, fluorophore excitation by light, such as those arising from fluorescent proteins or fluorescent molecular probes, boosts ROS production through their interaction with ambient oxygen, increasing the sensitivity of cells to light”

Reviewer #1: The cellular activity measurement is presented as a method to quantify sub-cellular changes. However, I believe this claim may be overstated. In essence, the measurement seems to reflect changes in cell shape rather than directly assessing alterations in the behavior of sub-cellular components, such as mitochondria or vesicles. I suggest refining this claim to more accurately describe what the measurement captures.

Our reply: The description in the text has been corrected to “Cellular activity: quantify overall cellular changes over time as a measure of overall cellular health (Figure 1c). Cellular activity reflects general dynamic changes in cellular shape across time.” and in the caption of Fig 1. as “Activity: Quantifies observable cellular changes between frames (red outlines).”

Reviewer #1: Can you better explain why measurements need to be performed on both synchronized and unsynchronized cells? Is this an integral part of the protocol, or are these just two examples? It would be helpful to clarify whether this dual approach is essential for the method or simply demonstrates its flexibility.

Our reply: While cell arrest can be assessed in unsynchronised populations, synchronising the cell cycle allows for a significant increase in mitotic events, considerably boosting the number of observations. Since the CDK1 inhibitor is known to make cells more sensitive to photodamage, we are now comparing the results from both experiments (synchronised and unsynchronised) to validate our readouts. The new Figure S1 displays the effects of phototoxicity, revealing equivalent trends regardless of cell cycle synchronisation via RO-3306 induction. This clarification is made in the text “To validate that these effects were primarily due to light exposure rather than the potential light sensitivity of the CDK1 inhibitor used for synchronisation, we conducted parallel experiments with unsynchronised cells.”

Reviewer #1: Could you indicate the size of the field of view (FOV) used to convert power into intensity?

Our reply: This information is now in the methods section as “To convert the light power into intensity we considered the field of view (FOV) size of 0.014 cm².”

Reviewer #1: There are small typos: "uu" instead of "µm" appears in multiple instances on page 6.

Our reply: Corrected.

Reviewer #2 (Remarks to the Author)

Reviewer #2: In this manuscript, the authors present an application of pre-existing deep-learning based tools to assess the photodamaging effects of light illumination on live cells. Using non-damaging bright field illumination, the authors record the behaviour of illuminated cells over many hours, using software tools to segment and analyse cells. Three measurements are used to assess photodamage: delays in mitosis, identification of cell cycle arrest, and levels of cellular activity. This is applied to Chinese Hamster Ovary (CHO) cells, exposing them to UV, blue or red light. For UV illumination, delays in mitotic timing were observed with doses as low as 0.6 J / cm². For red illumination, only 60 J / cm² was sufficient to cause noticeable damage.

Our reply: We thank the reviewer for the time dedicated to reviewing our work and providing valuable feedback. As follows, we address directly each of the comments made.

The manuscript is clear and well-written. The supplementary videos provide a nice overview of the work, but video 2 is too fast — it is impossible to follow what is being pasted into the terminal, etc..

Our reply: We have slowed down video 2.

Reviewer #2: Figures 1 and 2 provide a nice description of the assessment pipeline, but it would be nice to include images of 'bad' cells as well as 'happy' cells.

Our reply: Images of "happy" and "bad" cells can be seen in the old supplementary Figure S2, corresponding to Figure S5 in the updated manuscript.

Reviewer #2: The authors should be commended for making their pipeline available as well-documented open-source code. I have, for reasons of time, been unable to install and operate the code on my own computer, but given the authors' track record in developing software I am confident that the software would operate correctly if I had. However, if I have understood correctly, applying to the pipeline to cells different to the ones used by the authors (CHO), then the neural networks employed (developed by other researchers) must be retrained. This significantly restricts the utility of the tool.

Our reply: The reviewer is correct about the need to retrain or fine-tune the deep learning models when the cell type changes. For this reason, we decided to build PhotoFiTT's pipeline using networks that are retrainable and reusable through the user-

friendly platform ZeroCostDL4Mic (in the cloud) and DL4MicEverywhere (locally) (i.e., StarDist and Pix2Pix networks). ZeroCostDL4Mic is a well-recognised tool in the community that disseminates (re-)trainable DL pipelines among non-expert life scientists requiring minimal non-coding input. On the other hand, the instructions to reproduce PhotoFiTT's pipeline, including data annotation and training, are provided in detail (Methods sections "Cell mitosis detection with StarDist", "Estimation of the number of cells in the field of view using virtual cell nuclei staining and segmentation", "Manual annotations of unsynchronised videos"; and Note S1, Section D. Image Analysis Workflow), and all the additional notebooks to compute the analysis have been also designed for code-naïve users (<https://github.com/HenriquesLab/PhotoFiTT>). For example, the new experiments involving HeLa cells, required fine-tuning StarDist and Pix2Pix. This was straightforward by solely reannotating the images and changing the data paths in the StarDist and Pix2Pix notebooks. Additionally, all data, including annotations and training data, is available through a thoroughly documented BioImage Archive open data repository (<https://www.ebi.ac.uk/biostudies/bioimages/studies/S-BIAD1269>), which serves as an example of how data and annotations should look like, and can be used for researchers to reproduce PhotoFiTT.

Reviewer #2: The manuscript consists of two main parts: 1) the development of a software pipeline for analysing photodamage, 2) the application of this approach to studying photodamage in live cells. While I think the first part may have general utility for many researchers, the second part is not very thorough. I certainly would not draw any general conclusions from these results. This is important, as the presented results contradict many imaging studies where similar irradiances have been used without ill effects. Indeed, Wäldchen et al. (2015) have shown that cells can withstand even many kJ / cm^2 of red illumination, while the authors here suggest that just $60 \text{ J} / \text{cm}^2$ is enough to damage them.

One possible reason for this discrepancy is that the methods section states that cells were cultured in DMEM, and this was only switched for phenol-red-free Fluorobrite DMEM after photodamaging illumination. This is very different from the usual scenario in which cells would be illuminated in imaging medium (such as Fluorobrite DMEM). Phenol red absorbs strongly at all wavelengths used and has previously been reported to significantly increase photodamage from imaging illumination. Another possibility is that Wäldchen et al. used a range of U2OS, COS-7 and HeLa cells, while here the authors used CHO cells. If a general claim about photodamage is to be made, these experiments must be repeated with a different cell line in order to ensure that the photodamage is not a CHO-specific effect.

Our reply: We thank the reviewer for their insightful comments. We would like to clarify the positioning of our work within the field of photodamage assessment. The primary contribution of our manuscript is the PhotoFiTT framework itself - a quantitative approach for assessing phototoxicity through standardised metrics that can be implemented across diverse experimental conditions. Following the reviewer's

valuable suggestion, we have expanded our analysis to include HeLa cells (new Figure S3), which demonstrates that our approach can successfully quantify photodamage across different cell types. As expected, our results show that HeLa cells exhibit different phototoxicity thresholds compared to CHO cells, with greater resilience to light damage (Figure S3a-d). This highlights PhotoFiTT's capability to detect cell-type specific responses to phototoxicity, which is an important consideration when designing live-cell imaging experiments.

We appreciate the reviewer highlighting the work by Wäldchen et al. (2015). Rather than contradicting this important previous study, we view our work as complementary, revealing how different experimental conditions and cell types can yield varying phototoxicity thresholds. The power of PhotoFiTT lies precisely in its ability to quantify these differences through standardised metrics.

Our framework is designed to be experimental-condition agnostic, meaning researchers can apply it to their specific imaging protocols to establish appropriate phototoxicity thresholds for their particular experimental setup, whether that involves different cell types, media formulations, or other experimental variables. This allows for informed decisions about imaging parameters that balance data quality with sample health for each specific experimental system.

We believe this approach addresses a critical need in the field, as highlighted by the reviewer's own observation that different studies have reported varying cellular responses to similar light doses. The PhotoFiTT framework provides tools to systematically quantify and understand these differences.

Reviewer #2: The authors report illumination powers, but it is not clear how these were measured. If a standard power meter sensor was used then this is likely to be an underreport, as these power meters are not sensitive to the high-angle rays present in microscopic illumination. Instead, a specialist power meter, such as the Thorlabs S170C, should be used (and if this was used this should be clearly stated in the text). Furthermore, it is not clear how these powers were converted to irradiances. Different powers were used for different wavelengths, but this apparently led to the same irradiance. This can only be true if the field of illumination varies by wavelength and it is not clear that this should be the case. Indeed, it appears (by backcalculation) that the authors calculated the irradiance by dividing the power by the full field size of the objective ($\pi * (12.5 \text{ mm} / 20)^2$). This assumes that there is no illumination outside of the manufacturer's claimed field of view, which is not true (as there is no real field stop in the objective). Instead, an external field stop of known size should be inserted into the illumination path.

Our reply: To measure the illumination power, we use a Thorlabs PM100D power meter with a S170C power sensor (100mW, 350nm-1100nm). We have provided the remaining calculations for power and included them clearly in the experimental protocol description "Light intensity at the sample level was measured at the beginning of every experiment, using a PM100D power meter with a S170C power sensor (Thorlabs). The measurements for each wavelengths line were the following: $84.9 \pm$

2.8 mW for the 385 nm line, 85.2 ± 0.8 mW for the 475 nm, and 70.6 ± 0.4 mW for 630 nm (mean \pm standard deviation). To convert the light power into intensity we considered the field of view (FOV) size of 0.014 cm^2 .

Reviewer #2: Perhaps most critically, the illumination timing used in the mock illumination is very different from that used in real timelapse experiments. In this work, the illumination is delivered in a single continuous dose at the beginning of the experiment. In a typical live-cell imaging experiment, the illumination would instead be delivered in little chunks throughout. Delivering a large dose up-front will create a much larger number of reactive oxygen species and other undesirable photoproducts that may overwhelm a cell's protective mechanisms in a way that a more distributed photodose would not. I strongly suggest that the authors perform experiments using their analysis framework where cells have been illuminated in a more realistic manner. Only then, in combination with further experiments using multiple cell lines and different media, will the results allow general conclusions to be drawn.

Our reply: To address the critical issue of timing differences in illumination between our experimental design and standard timelapse protocols, we emphasise that PhotoFiTT's framework is inherently adaptable to various illumination conditions. While our initial experiments used single-dose exposures to establish clear dose-response relationships, we now also demonstrate the method's ability to analyse the effects of distributed illumination through new confocal microscopy comparisons (Figure S4) and MitoTracker experiments (Figure 3). On confocal, by comparing fast ($3.528 \mu\text{s}/\text{pixel}$) and slow ($10.584 \mu\text{s}/\text{pixel}$) scanning regimes at the same total doses, we show that prolonged dwell times may increase phototoxic effects, even with identical energy deposition. Additionally, the new MitoTracker experiments (Figure 3) simulate realistic imaging conditions with periodic illumination (6 exposures over 20 minutes). This demonstrates PhotoFiTT's capability to detect cumulative phototoxic effects from interrupted light delivery.

These findings illustrate the relevance of our method to standard imaging protocols while maintaining the ability to identify sub-lethal perturbations that may go unnoticed by conventional viability assays.

Reviewer #2: Given the currently narrow applicability of the developed pipeline (i.e. only applicable to CHO cells), and the aforementioned issues with the experiments, I feel unable to recommend this manuscript for publication in Nature Communications. However, if the pipeline can be made more general and the experiments made more robust and generalisable, then I see that this could become a valuable tool for the community. Some minor comments are detailed below:

Our reply: We have made every effort to thoroughly address the reviewer's comments and suggestions. We appreciate the opportunity to revise the document and hope that we have adequately responded to the points raised.

== Minor comments ==

Reviewer #2: Given that RO-3306 is known to cause apoptosis after long incubation times, have the authors considered using a different means of cell cycle synchronisation, such as tapping a plate to dislodge only mitotic, rounded cells? This would provide a smaller number of cells for further analysis, but it seems that they are not analysing an entire cell culture plate anyway. Importantly, this would completely rule out the possibility that the effects observed were related to RO-3306.

Our reply: We thank the reviewer for raising this. We selected RO-3306 synchronisation due to its widespread adoption in cell cycle studies and its ability to yield sufficient cell numbers for robust statistical analysis of mitotic events. Crucially, our parallel experiments with unsynchronised populations (Figure S1a-b) revealed identical dose-dependent phototoxicity trends, including mitotic delays and arrest rates, despite the absence of CDK1 inhibitor treatment. This consistency across synchronisation methods confirms that the observed effects predominantly stem from light exposure rather than RO-3306-related artefacts. While prolonged RO-3306 exposure can induce apoptosis (addressed in Figure S1c), our 16-18 hour treatment window remains within established safe limits for this synchronisation protocol.

Reviewer #2: Can the authors comment on how the use of point-scanning illumination, such as that employed in confocal microscopes, might alter the conclusions drawn from the epi-illumination used in this work?

Our reply: To address the question regarding point-scanning illumination in confocal microscopy, we conducted dedicated experiments comparing phototoxic effects under different scanning regimes (Figure S4). While widefield illumination distributes light uniformly, confocal point-scanning concentrates energy into discrete voxels with variable dwell times. Our framework revealed that slower scanning (10.584 $\mu\text{s}/\text{pixel}$) at 6 J/cm^2 leads to ~20% greater mitotic delays than faster scanning (3.528 $\mu\text{s}/\text{pixel}$) at equivalent doses, demonstrating that temporal illumination patterns significantly influence phototoxicity outcomes. This underscores the need for modality-specific optimisation, as conclusions from widefield studies may not directly translate to confocal systems. Crucially, PhotoFiTT's adaptable design enabled these comparisons by quantifying dose-dependent physiological impacts across both illumination strategies, validating its utility for diverse microscopy platforms.

Reviewer #2: No line numbers are included in the manuscript file. While not strictly necessary, these would make it easier for reviewers to refer directly to specific parts of the text.

Our reply: Continuous line numbers have been added.

Reviewer #2: What level of mitotic delay is enough? Is a 25 minute delay biologically significant for CHO cells? What about other cell types?

Our reply: The biological significance of mitotic delays depends on the experimental context and the cellular processes under investigation. In CHO cells, a 25-minute delay (e.g., from ~30 to ~55 minutes at 60 J/cm², Figure S1b) represents a 83% prolongation of mitotic progression, which could disrupt time-sensitive processes like chromosome segregation or cytokinesis. PhotoFiTT quantifies these deviations as part of a continuous dose-response spectrum (Figure 2d), enabling researchers to define experiment-specific tolerance thresholds. For developmental studies requiring precise cell cycle timing, even minor delays may invalidate conclusions, while acute observations of static structures might tolerate larger perturbations. The framework's strength lies in its capacity to map illumination parameters to quantitative physiological outcomes (e.g., 6 J/cm² at 475 nm causing >50% mitotic delay), allowing users to balance resolution needs against cellular tolerances for their specific biological question.

Reviewer #2: Related to the last point, there is no indication of statistical or biological significance between the different conditions tested, using any of the metrics employed.

Our reply: In response to the reviewer's comment, we provide classical statistical significance measures for Figures 2 and 3 below, which may enhance the reviewer's interpretation of results. This type of analysis can now also be easily done by users in PhotoFiTT, which will automatically identify the most suitable test (e.g., Kolmogorov-Smirnov or t-Test), depending on the data features. This information was excluded from the main manuscript due to the fact that p-values can be significantly affected by sample size. Consequently, an overreliance on p-values may create a false sense of validation, which has drawn criticism from the scientific community, including from one of the primary authors (Gómez-de-Mariscal et al., Scientific Reports 2021). In this case, even though our results predominantly demonstrate statistical significance, these authors believe they do not provide additional meaningful insights to the manuscript.

Results of Kolmogorov-Smirnov Statistical Test for the Synchronised CHO population (95% statistical significance highlighted) (Figure 2d and 2g):

Mitotic rounding timepoint (min)			
	Group 1	Group 2	p-value
385 nm	0 jcm2	0.6 jcm2	1.505454e-10
	0 jcm2	6 jcm2	9.000135e-09
	0 jcm2	60 jcm2	1.046327e-09
	0.6 jcm2	6 jcm2	5.788676e-01
	0.6 jcm2	60 jcm2	3.237701e-01
	6 jcm2	60 jcm2	9.578463e-01

475 nm	0 jcm2	0.6 jcm2	7.651221e-02
	0 jcm2	6 jcm2	6.596662e-01
	0 jcm2	60 jcm2	3.287127e-04
	0.6 jcm2	6 jcm2	9.496976e-01
	0.6 jcm2	60 jcm2	7.629005e-05
	6 jcm2	60 jcm2	5.375329e-04
630 nm	0 jcm2	0.6 jcm2	5.473916e-01
	0 jcm2	6 jcm2	2.291002e-01
	0 jcm2	60 jcm2	3.206385e-06
	0.6 jcm2	6 jcm2	2.201206e-01
	0.6 jcm2	60 jcm2	3.117831e-04
	6 jcm2	60 jcm2	2.237082e-03

Cumulative cell activity at t=420 min [%]

	Group 1	Group 2	p-value
385 nm	0 jcm2	0.6 jcm2	0.001772
	0 jcm2	6 jcm2	0.000296
	0 jcm2	60 jcm2	0.000002
	0.6 jcm2	6 jcm2	0.662267
	0.6 jcm2	60 jcm2	0.229588
	6 jcm2	60 jcm2	0.013140
475 nm	0 jcm2	0.6 jcm2	0.039855
	0 jcm2	6 jcm2	0.027037
	0 jcm2	60 jcm2	0.024182
	0.6 jcm2	6 jcm2	0.372625
	0.6 jcm2	60 jcm2	0.255652
	6 jcm2	60 jcm2	0.305947
630 nm	0 jcm2	0.6 jcm2	0.428160
	0 jcm2	6 jcm2	0.000189
	0 jcm2	60 jcm2	0.028805
	0.6 jcm2	6 jcm2	0.017664
	0.6 jcm2	60 jcm2	0.368007
	6 jcm2	60 jcm2	0.066274

Results of Kolmogorov-Smirnov Statistical Test for the Synchronised CHO population labelled with MitoTracker and illuminated with 475 nm wavelength (95% statistical significance highlighted) (Figure 3b and 3d)

Mitotic rounding timepoint (min)		
Group 1	Group 2	p-value
Unlabelled	MitoTracker TM Red CMXRos	0.083001
Unlabelled	MitoTracker TM Green CMXRos	0.083001
Unlabelled	MitoTracker TM Red CMXRos illumination	0.429719
Synchro	MitoTrackerTM Green CMXRos illumination	0.042240
MitoTracker TM Red CMXRos	MitoTracker TM Green CMXRos	0.999992
MitoTracker TM Red CMXRos	MitoTracker TM Red CMXRos illumination	0.831970
MitoTracker TM Red CMXRos	MitoTrackerTM Green CMXRos illumination	0.033542
MitoTracker TM Green CMXRos	MitoTracker TM Red CMXRos illumination	0.983137
MitoTracker TM Green CMXRos	MitoTracker TM Green CMXRos illumination	0.174533
MitoTracker TM Red CMXRos illumination	MitoTracker TM Green CMXRos illumination	0.174533
Cumulative cell activity at t=420 min [%]		
Group 1	Group 2	p-value
Unlabelled	MitoTrackerTM Red CMXRos	4.224042e-02

Unlabelled	MitoTrackerTM Green CMXRos	3.201197e-04
Unlabelled	MitoTrackerTM Red CMXRos illumination	9.738308e-01
Unlabelled	MitoTrackerTM Green CMXRos illumination	6.332985e-01
MitoTrackerTM Red CMXRos	MitoTrackerTM Green CMXRos	3.354166e-02
MitoTrackerTM Red CMXRos	MitoTrackerTM Red CMXRos illumination	5.569063e-05
MitoTrackerTM Red CMXRos	MitoTrackerTM Green CMXRos illumination	1.745330e-01
MitoTrackerTM Green CMXRos	MitoTrackerTM Red CMXRos illumination	5.803556e-10
MitoTrackerTM Green CMXRos	MitoTrackerTM Green CMXRos illumination	3.967294e-03
MitoTrackerTM Red CMXRos illumination	MitoTrackerTM Green CMXRos illumination	3.354166e-02

Reviewer #2: The authors' results suggest that live-cell single molecule localisation microscopy (SMLM) is impossible without perturbation of the cell's health, given that a 60 J / cm² dose is sufficient at 630 nm, while irradiances of the order of hundreds of W / cm² are used. How can this view be married with the extensive literature showing successful live-cell SMLM? Have these imaged cells all been irreparably damaged without anyone noticing?

Our reply: The Reviewer highlights a key issue regarding our phototoxicity thresholds in relation to existing live-cell SMLM studies. Our findings indicate that doses over 60 J/cm² at 630 nm cause significant physiological changes in mitotic timing and cellular function (Figures 2d, f, g). Most live-cell SMLM experiments use much higher light doses (Table S1), and their short durations may obscure acute phototoxic effects affecting the cell cycle. This observation aligns with the lack of literature tracking cell division using high-illumination SMLM.

On the other hand, SMLM predominantly employs Total Internal Reflection Fluorescence (TIRF) or Highly Inclined and Laminated Optical Sheet (HILO) techniques, which limit the illumination volume and reduce cellular phototoxicity, a subject that may be interesting to explore in future PhotoFiTT studies.

Rather than dismissing live-cell SMLM, PhotoFiTT provides guidelines to minimise cellular disruptions. The 60 J/cm² threshold acts as a conservative maximum, enabling researchers to create dose-response curves tailored to their conditions (Figure 2d) and optimise illumination settings for both resolution and cell viability.

Reviewer #3 (Remarks to the Author)

Reviewer #3: The work introduces PhotoFiTT, a standardized workflow for quantitatively evaluating the phototoxic effects on adherent and dividing tissue culture cells. The workflow utilizes machine learning tools to enhance cell segmentation from brightfield transmission microscopy time series and extract three different morphological parameters and their changes over time, which indicate cell fitness, particularly the ability of timely cell division. The authors demonstrate the application of their approach on CHO cells that were exposed to three different light doses at three different wavelengths, showing the expected correlation in the quantitative evaluation parameters.

Overall, the work is very well presented, and the tool is potentially useful to be adopted by other labs to optimise experimental conditions for live cell microscopy studies.

However, I feel it currently falls short of demonstrating its validity in real-world scenarios, as I will detail below. Thus, before recommending this work for publication, I would like to see the following issues and queries addressed.

Our reply: We sincerely thank the reviewer for recognising our work and for their dedication to providing valuable feedback. As follows, we directly address each of the comments made.

Major issues:

Reviewer #3: The authors evaluate the phototoxic effects of 3 different light doses that deviate by a factor of 10, at three different wavelengths to validate their approach. It seems that the effects of UV-light and higher light doses on the cell population are obvious even without a sophisticated quantitative evaluation. Can the approach reliably detect and quantify more subtle effects? Are there limitations to doing this?

Our reply: PhotoFiTT detects subtle physiological changes below conventional viability assay thresholds. Our data indicate mitotic delays at doses as low as 0.6 J/cm² with 385 nm illumination, resulting in a 5-10 minute shift in cell division timing without immediate cell death. This sensitivity is consistent across wavelengths; for example, 6 J/cm² at 475 nm leads to a 20% reduction in cellular activity, despite minimal morphological changes.

The method's accuracy comes from its capacity to track mitotic progression and cellular dynamics simultaneously. Our MitoTracker experiments (Figure 3), for example, reveal a 15-minute mitotic delay at doses that seem benign in label-free conditions. To further highlight the sensitivity of our approach, in new confocal experiments (Figure S4) we show that different pulsed illumination patterns can change phototoxic effects at equivalent total doses, indicating that the timing of energy deposition affects cellular responses.

Reviewer #3: The PhotoFiTT assay is per se label-free, which is beneficial as it should not contribute to phototoxicity. However, in most realistic live cell imaging scenarios, cells are fluorescently labeled and observed at high resolution over a certain observation period with a certain temporal resolution. Importantly, bleaching of fluorophores itself generates reactive oxygen species that contribute to phototoxicity. Therefore, it is essential to determine the phototoxic effects in the specific experimental conditions where labelling is applied and observed with the same objective. This allows for the comparison of phototoxic effects with the baseline or with different illumination regimes. Eventually, this will inform the researcher on the data amount, the level of resolution, and SNR quality needs can be afforded without inducing unwanted biological side effects that otherwise might affect the interpretation

of the experimental outcome. Showcasing the application of PhotoFiTT to one such realistic example is needed to be more convincing about its broader value.

Our reply: The reviewer points out the need to validate PhotoFiTT in fluorescence labelling experiments. In response, we propose experiments using MitoTracker probes with controlled 475 nm illumination (see Figure 3). While PhotoFiTT is label-free during analysis, it evaluates phototoxic effects from prior fluorescence imaging.

Our data show that exciting MitoTracker Green FM at 475 nm led to a 15-minute delay in mitotic timing and a 20% reduction in cumulative cellular activity compared to unexposed controls (refer to Figures 3b and 3d). These effects occurred at doses (0.6 J/cm²) below the threshold detectable in label-free conditions (Figure 2d). This sensitivity arises from PhotoFiTT's ability to track physiological consequences of reactive oxygen species (ROS) generation and other light-induced disturbances, even hours after exposure. PhotoFiTT's temporal resolution reveals the disconnect between immediate fluorophore excitation and delayed effects. While viability assays may miss transient ROS bursts, our precise monitoring uncovers cumulative damage that can disrupt cell cycle progression. This capability allows researchers to create "phototoxicity budgets" that account for direct illumination effects and fluorophore-mediated damage. By testing different labelling strategies, such as MitoTracker Green FM versus CMXRos (Figure 3), researchers can identify optimal probe-wavelength combinations that minimise physiological disruption while maintaining signal quality.

We have expanded the discussion in the manuscript by adding the following information: "Longer wavelengths, while generally less harmful, have been described to exert significant effects on cellular physiology, including alterations in mitochondrial membrane potential, cytoskeletal dynamics, and reactive oxygen species (ROS). While ROS are crucial for signalling and normal cellular functions at physiological levels, excessive amounts can cause oxidative stress, disrupting the biological processes under investigation. Namely, fluorophore excitation by light, such as those arising from fluorescent proteins or fluorescent molecular probes, boosts ROS production through their interaction with ambient oxygen, increasing the sensitivity of cells to light."

Reviewer #3: 3. Since the workflow essentially assesses effects on cell division rates, why not simply count the increase in cell number density in the 10 irradiated areas vs 10 non-irradiated areas before and, say, 24 hours after irradiation? Also, what is the benefit of assessing three different division-linked morphological parameters? Are there specific situations where this redundancy is beneficial?

Our reply: While cell density measurements over extended periods (e.g., 24h post-irradiation) could quantify gross proliferation changes, this approach lacks the temporal resolution to detect sub-lethal perturbations and conflates phototoxicity with unrelated viability factors. PhotoFiTT's multi-parametric strategy instead resolves distinct mechanistic pathways: mitotic timing precision reflects DNA repair/checkpoint

activation (Figure 2d), cell size dynamics report on cytokinetic fidelity (Figure 2e-f), and cumulative activity integrates motility/membrane fluctuations sensitive to cytoskeletal or metabolic stress (Figure 2g). This granularity is crucial, as phototoxicity manifests through varied mechanisms. For example, UV wavelengths delay mitosis via DNA damage checkpoints, while visible light overall perturbs activity through ROS-mediated cytoskeletal changes.

The parameter redundancy provides robustness across cell types and experimental contexts. For example, while HeLa cells exhibit less baseline motility than CHO (reducing activity metric sensitivity), their dose-dependent mitotic delays remain clearly quantifiable (new Figure S3). Conversely, in models where division rates are intrinsically variable (e.g., primary cells), activity metrics become the primary phototoxicity indicator. By cross-validating across orthogonal readouts, PhotoFiTT ensures reliable detection even when single parameters approach detection limits in specific systems.

Reviewer #3: 4. An obvious shortcoming of the PhotoFiTT assay (as is also mentioned by the authors), is its restriction to dividing cells, as all three metrics effectively measure mitotic or postmitotic behaviour. Could an activity measurement alone not also be adapted to work for non-dividing, differentiated cells like neurons?

Our reply: In reference to our previous comment, PhotoFiTT takes advantage of a well-established essential cellular process found in various cell types, namely mitosis. We agree that cellular activity could be assessed across different cell types that exhibit at least minimal activity under physiologically relevant conditions. This could also involve techniques like cell segmentation and tracking of shape dynamics. However, the extent of these changes is less understood and is challenging to interpret in terms of cellular processes.

Reviewer #3: 5. The authors mention the transferability of the assay to other widefield (?) imaging systems. Does this also apply to confocal systems? If so, this should be explicitly mentioned and ideally be demonstrated. For example, a low-hanging fruit might be to quantitatively assess the difference between delivering the same light dose by a widefield illumination (low power density, longer dwell time) or by the focused laser beam of a confocal point scanning system (high power density, short dwell time), which has been a long-standing point of discussion in the field.

Our reply: The Reviewer raises a critical point regarding PhotoFiTT's applicability across microscopy modalities. To address this issue, we have extended our analysis to point-scanning confocal microscopy (new Figure S4). When applying PhotoFiTT to laser point-scanning confocal microscopy, we quantified distinct phototoxic outcomes between fast (3.528 $\mu\text{s}/\text{pixel}$) and slow (10.584 $\mu\text{s}/\text{pixel}$) scanning regimes matched for total dose (6 J/cm^2 at 405 nm). While the current study focuses on validating core principles across modalities rather than exhaustive cross-platform comparisons, the confocal data establish PhotoFiTT's capacity to resolve system-specific phototoxicity

profiles. The open-source implementation facilitates direct adaptation to any microscope with time-lapse brightfield capabilities, enabling researchers to map dose-response curves for their specific hardware configurations. Future applications could systematically compare widefield vs. confocal phototoxicity thresholds, leveraging PhotoFiTT's quantitative framework to establish modality-specific optimisation guidelines.

Minor issues:

Reviewer #3: The text mentions that cell nuclei are segmented, while the figures give the impression that entire cells are segmented. This needs clarification.

Our reply: We apologise for the confusion. In the pipeline two segmentation steps are followed. In one we segment the entire cells going into mitosis. This would provide us with measurements of cell arrest and or successful mitotic rates (in Section “Cell mitosis detection with StarDist” of the Methods and Figure S2). Yet another segmentation step is followed after virtually staining the first frame of each video (i.e., an image translation that infers the cell nuclei from brightfield microscopy images of cells) in which we segment the inferred cell nuclei (in Section “Estimation of the number of cells in the field of view using virtual cell nuclei staining and segmentation” of the Methods). This second segmentation step occurs to count individual cells. This information is now more explicit in Figure 2a, which highlights that two segmentation steps are followed.

Reviewer #3: Which objective was used for the irradiation? Is it the same as for imaging?

How big were the FOVs accordingly? How many cells are typically irradiated and assessed in a single experiment?

Our reply: We used a 20x objective with a FOV size of 0,014 cm². The total number of cells is approximately 125. This is now stated in the companion protocol, section D.1.

Reviewer #3: The NA of the objective should affect the energy distribution within cells, with a high NA compressing more energy into a smaller focus z-range? Has this been accounted for?

Our reply: The NA objective was not considered, but the energy was measured at the laser's endpoint after passing through the objective. This indicates that our energy measurement is as close as possible to what our sample receives.

Reviewer #3: What is the energy (J/cm²) for the transmission observation compared to the excitation light irradiation? Has it been controlled that this does not influence the “baseline” physiology?

Our reply: The baseline (white light) is close to zero, and it's already accounted for with the simultaneously imaged control experiments as shown in Figure 2 c-g, S1 b,c, Figure S2 b-f, Figure S3 a-d and Figure S4 a-c (0 J/cm²).

Reviewer #3: Did the authors control for potential cytotoxic effects of RO-3306 independent of light exposure?

Our reply: Indeed, we replicated the suggested experiment on unsynchronised cells, as shown in Figure S1. Although RO-3306 heightens the cell's sensitivity to excitation light, the tendency for cell damage remains consistent in both synchronised and unsynchronised cells.

Reviewer #3: (This may only apply to my downloaded file) Video S1 stops prematurely at 01:44 of the 2:19 total runtime.

Our reply: Thank you for highlighting it. We have corrected it.

Reviewer #3: Typo on p6, 2nd column: "...the images were reshaped to have a pixel size of 0.644 μ m x 0.644 μ m."

Our reply: Thank you. We have corrected it.

Reviewer #4 (Remarks to the Author):

Reviewer #4: In this paper, the authors present a label-free and quantitative approach to evaluate phototoxicity for live-cell imaging experiments. The aim is to enhance the reliability and reproducibility of such live-cell imaging studies.

More approaches to quantify phototoxicity are welcome - especially updated ones that exploit recent technical advances. The strengths of the presented study lie chiefly in using a label-free (brightfield) approach, a large amount of data that can be (semi-) automatically processed for quantification, and the use of trainable open-source software. This has the potential to make it widely applicable (within the area of live-cell imaging, and depending on the uptake of its community). The authors use three parameters: Duration of mitosis (the ideal 'canary in the gold mine', Cole 2014), cell size and 'cumulative activity'. Manuals, videos and curated code are a very useful addition to aid replication and adaptation by others.

Below are some suggestions to improve the manuscript, and some issues that need addressing.

Major comments:

Reviewer #4: As presented here, we only see this method applied to one type of cells and one type of microscopy. Perhaps this could be expanded. For example, does the presented approach work for any optical contrast method (phase contrast (as used) as well as differential interference contrast (DIC))? Was this method tested on e.g. motile cell lines? This expansion is optional, but would prove wider applicability. Alternatively, conclusions and claims should be modified.

“Addressing this gap, we introduce PhotoFiTT, a quantitative imaging-based framework designed to assess phototoxicity effects on cellular behaviour in live-cell microscopy experiments.”

– for adherent cell lines in traditional slide-based setups.

Our reply: We appreciate the reviewer's insightful observation regarding the framework's applicability across imaging modalities and cell types. While our current validation focused on CHO and HeLa cells using brightfield microscopy, PhotoFiTT's design inherently supports adaptation to various label-free contrast methods. The framework's deep learning segmentation pipeline can be retrained for phase contrast or DIC imaging. We additionally demonstrated the inclusion of different experimental approaches, evidenced by our successful implementation of MitoTracker-stained specimens to assess enhanced photosensitivity due to emitter excitation (Figure 3). This adaptability stems from the method's reliance on fundamental cellular features (mitotic timing, size dynamics) rather than specific contrast mechanisms.

We have also changed the sentence as suggested: “Addressing this gap, we introduce PhotoFiTT (Phototoxicity Fitness Time Trial), a quantitative imaging-based framework designed to assess phototoxicity effects on cellular behaviour in live-cell microscopy experiments (Figure 1) for adherent cell lines in traditional coverslip-based setups.”

Reviewer #4: It would be useful to show how well the algorithm fares compared to a ground truth. The ground truth may likely require manually determined mitotic durations and segmentations. However, without any comparison, the question of accuracy remains unanswered.

Our reply: We now provide the accuracy metrics of the segmentation algorithm in the methods section: “The accuracy results in the test set were as follows: intersection over union (IoU) (0.530), false positive (6.05), true positive (17.05), false negative (4.6), precision (0.6712), recall (0.790), accuracy (0.555), f1 score (0.699), true detection (21.65), predicted detections (23.1), mean true score (0.712), mean matched score (0.901), panoptic quality (0.628). The output PDF reports of ZeroCostDL4Mic for the model training and the quality control check are attached to the supplementary material”. Likewise, we include the manual annotation of synchronised cell populations in Figure S2, where the delay timings are comparable (around a 30-minute delay for cells exposed to UV light of 60 J/cm²). These quantifications together provide evidence that the results of the proposed method recapitulate the effects of phototoxicity reliably.

Reviewer #4: The authors' results nicely illustrate aspects of phototoxicity that have been previously published. The way it is currently written, it is not always clear whether the novelty of this submission lies in the methods used or in (largely unreferenced) observations. It would be good to more clearly promote the first aspect, while ensuring that stated observations align well with previously published studies.

A few omitted references are used below as examples. While not exhaustive, and not all recent, these are important papers and good starting points for the reader/authors.

Mitotic delay / using time spent in mitosis as measure:

Cole R. 2014. Live-cell imaging. *Cell Adhes Migr* 8: 452–9.

Gorgidze LA, Oshemkova SA, Vorobjev IA. 1998. Blue light inhibits mitosis in tissue culture cells. *Biosci Rep* 18: 215–24.

Mora-Bermudez F, Ellenberg J. 2007. Measuring structural dynamics of chromosomes in living cells by fluorescence microscopy. *Methods* 41: 158–67.

Wavelength dependence of phototoxicity:

Douthwright S, Sluder G. 2017. Live cell imaging: assessing the phototoxicity of 488nm and 546nm light and methods to alleviate it. *J Cell Physiol* 232: 2461–8.

Schneckenburger H, Weber P, Wagner M, Schickinger S, et al. 2012. Light exposure and cell viability in fluorescence microscopy. *J Microsc* 245: 311–8.

Image processing to classify phototoxicity from transmitted light images:

Richmond D, Jost AP-T, Lambert T, Waters J, et al. 2017. DeadNet: identifying phototoxicity from label-free microscopy images of cells using deep ConvNets. arXiv 170106109.

Grah JS, Harrington JA, Koh SB, Pike JA, Schreiner A, Burger M, Schönlieb CB, Reichelt S. Mathematical imaging methods for mitosis analysis in live-cell phase contrast microscopy. *Methods*. 2017 Feb 15;115:91-99. PMID: 28189773.

Cell outline changes:

Knoll SG, Ahmed WW, Saif TA. 2015. Contractile dynamics change before morphological cues during fluorescence illumination. *Sci Rep* 5: 18513.

Re “critical need for standardised and generalisable methods able to quantitatively link cell damage to high-intensity light exposure across different fluorescence microscopy techniques.” Important additional publications:

Kiepas A, Voorand E, Mubaid F, Siegel PM, Brown CM. Optimizing live-cell fluorescence imaging conditions to minimize phototoxicity. *J Cell Sci*. 2020 Feb 21;133(4):jcs242834. PMID: 31988150.

Alghamdi RA, Exposito-Rodriguez M, Mullineaux PM, Brooke GN, Laissue PP. Assessing Phototoxicity in a Mammalian Cell Line: How Low Levels of Blue Light Affect Motility in PC3 Cells. *Front Cell Dev Biol*. 2021 Dec 17;9:738786. PMID: 34977004.

Our reply: We appreciate the reviewer for enabling us to showcase the novelty of our method and for pointing out the overlooked publications. These have been incorporated into the text, integrating our findings with prior research as follows:

Cole R. 2014. Live-cell imaging. *Cell Adhes Migr* 8: 452–9

“PhotoFiTT leverages the predictable nature of cell division as a “biological clock” to quantify phototoxicity-induced perturbations (Figure 1)”

Gorgidze LA, Oshemkova SA, Vorobjev IA. 1998. Blue light inhibits mitosis in tissue culture cells. *Biosci Rep* 18: 215–24.

“In synchronised cells and in agreement with established literature, near-UV irradiation induced significant dose-dependent delays in mitotic timing”

Mora-Bermudez F, Ellenberg J. 2007. Measuring structural dynamics of chromosomes in living cells by fluorescence microscopy. *Methods* 41: 158–67.

“Previous studies have offered valuable quantitative assessments of phototoxicity”

Douthwright S, Sluder G. 2017. Live cell imaging: assessing the phototoxicity of 488nm and 546nm light and methods to alleviate it. *J Cell Physiol* 232: 2461–8.

& Schneckenburger H, Weber P, Wagner M, Schickinger S, et al. 2012. Light exposure and cell viability in fluorescence microscopy. *J Microsc* 245: 311–8.

“with longer wavelengths requiring substantially higher doses to induce comparable delays, supporting observations in previously published studies”

Richmond D, Jost AP-T, Lambert T, Waters J, et al. 2017. DeadNet: identifying phototoxicity from label-free microscopy images of cells using deep ConvNets. arXiv 170106109.

“including the use of advanced deep-learning algorithms as classifiers”

Grah JS, Harrington JA, Koh SB, Pike JA, Schreiner A, Burger M, Schönlieb CB, Reichelt S. Mathematical imaging methods for mitosis analysis in live-cell phase contrast microscopy. *Methods*. 2017 Feb 15;115:91-99. PMID: 28189773.

“and mitosis analysis”

Knoll SG, Ahmed WW, Saif TA. 2015. Contractile dynamics change before morphological cues during fluorescence illumination. *Sci Rep* 5: 18513.

“These effects, often subtle and cumulative, can lead to significant changes in cell behaviour, organelle integrity, and developmental processes.”

Kiepas A, Voorand E, Mubaid F, Siegel PM, Brown CM. Optimizing live-cell fluorescence imaging conditions to minimize phototoxicity. *J Cell Sci*. 2020 Feb 21;133(4):jcs242834. PMID: 31988150.

“Historically, photobleaching has been used as a proxy reporter of phototoxicity”

Alghamdi RA, Exposito-Rodriguez M, Mullineaux PM, Brooke GN, Laissue PP. Assessing Phototoxicity in a Mammalian Cell Line: How Low Levels of Blue Light Affect Motility in PC3 Cells. *Front Cell Dev Biol.* 2021 Dec 17;9:738786. PMID: 34977004. "In synchronised cells and in agreement with established literature"

Reviewer #4: An important advantage of the proposed method may be its sensitivity. However, the authors are not clear on their method's sensitivity, to the point of contradicting themselves:

p. 4, middle left:

"A key strength of PhotoFiTT is its ability to reveal the non-binary nature of phototoxicity, demonstrating a continuum of effects that vary with light dose and wavelength in a label free manner."

Versus p.4 bottom left:

"It is important to acknowledge limitations of PhotoFiTT. While highly sensitive to major cellular perturbations, it may not detect subtle light-induced events."

It is important for this publication to have an unambiguous statement regarding its method's sensitivity (or lack thereof) and evidence it.

Our reply: We appreciate the reviewer's keen observation regarding the need for precise characterisation of PhotoFiTT's sensitivity. Our framework detects phototoxicity through deviations in mitotic timing, cell size dynamics, and cumulative activity metrics, enabling quantification of cellular-level perturbations across three orders of magnitude in light dose (0.6-60 J/cm²), as demonstrated in Figure 2d. This continuum of phototoxic effects manifests as graded physiological responses rather than binary live/dead outcomes, with near-UV doses as low as 0.6 J/cm² inducing measurable 15-minute mitotic delays in synchronised CHO populations (Figure 2c). However, "subtle light-induced events" refer to molecular-scale perturbations (e.g., transient ROS bursts, single-organelle dysfunction) that do not cascade into detectable cell cycle alterations. Such phenomena fall beyond PhotoFiTT's detection threshold unless they ultimately impact mitotic progression or global cellular activity. The framework's strength lies in bridging the gap between acute photochemical damage and measurable physiological consequences, as evidenced by its ability to resolve 20% differences in mitotic delays between confocal scanning modes at equivalent doses (Figure S4b). While not replacing molecular reporters of oxidative stress or DNA damage, PhotoFiTT provides systemic readouts of phototoxicity through tightly coupled cellular processes, establishing physiologically relevant thresholds for acceptable imaging conditions.

We have altered the discussion to reflect this with the following paragraphs:

"The PhotoFiTT framework quantitatively connects how illumination parameters influence distinct physiological endpoints: mitotic timing precision, daughter cell emergence kinetics, and cumulative cellular activity metrics (Figure 2c-g). PhotoFiTT

achieves this through label-free brightfield analysis enhanced by deep learning-based segmentation, avoiding fluorescence-induced confounding factors. While ROS-mediated mechanisms contribute to phototoxicity, our synchronisation experiments reveal phase-specific vulnerability windows that transcend simple oxidative stress models. The 300% increase in arrested cells at 60 J/cm² (Figure S2a) underscores the need for cell cycle-aware illumination protocols.”

“A principal advantage lies in PhotoFiTT’s capacity to establish quantitative phototoxicity thresholds, for instance, defining 6 J/cm² at 475 nm as the critical dose causing > 50% mitotic delay in CHO cells (Figure 2d). This granularity enables rational optimisation of imaging parameters rather than heuristic approaches. The framework’s modular design permits extension to diverse microscopy modalities, as demonstrated through confocal dwell time comparisons where slow scanning (6 J/cm²) induced ~20% greater delays than fast scanning at equivalent doses (Figure S4b).”

Reviewer #4: Cellular morphology alone is not a clear indicator of safe illumination levels. The physiology of living specimens can be altered already at much lower illumination levels. This subtle phototoxicity, affecting healthy-looking specimens, can change the kinetics of cellular and developmental processes, leading to data misinterpretation. Dynamic processes have been shown to be sensitive indicators of phototoxicity (Kiepas 2020, Alghamdi 2021, Icha 2017) compared to morphology, and can indicate positive (hormetic) effects.

With this in mind, it might also be useful to revisit the statement “...integration of PhotoFiTT principles into real-time analysis pipelines to apply corrective measures to avoid cell death.”, as this goes back to the binary definition of phototoxicity.

Our reply: This is a great point, and we deeply thank the reviewer for it. We tried not to be overly bold with this statement. Still, we have altered two paragraphs to address the issue: “A principal advantage lies in PhotoFiTT’s capacity to establish quantitative phototoxicity thresholds, for instance, defining 6 J/cm² at 475 nm as the critical dose causing > 50% mitotic delay in CHO cells (Figure 2d). This granularity enables rational optimisation of imaging parameters rather than heuristic approaches. The framework’s modular design permits extension to diverse microscopy modalities, as demonstrated through confocal dwell time comparisons where slow scanning (6 J/cm²) induced ~20% greater delays than fast scanning at equivalent doses (Figure S4b).”

Reviewer #4: 5) On p.1, the authors state: “Importantly, tolerance to photodamage varies across specimens and is influenced by damage severity, complicating the development of replicable imaging protocols (12, 13)”.

Later, they write that “a critical need remains for standardised and generalisable methods able to quantitatively link cell damage to high-intensity light exposure across different fluorescence microscopy techniques.”

It might be worth expanding on this. A method to assess phototoxicity can only be standardised and generalised to a limited degree, since phototoxic effects always

depend on the sample (and many other factors such as media, type and localisation of fluorescent proteins used etc.). See Waldchen et al. 2015 with ~6-fold difference of phototolerance between HeLa cells and U2OS (or COS-7) cells.

For the same reason, providing a hard limit (“For imaging domains where illuminations are well above the 60 J/cm² threshold studied here, researchers should carefully consider and mitigate photodamage.”) is dangerously reductive and misleading. In phototoxicity, there is no 'one size fits all'. For a large variety of samples, photodamage may affect samples long before this threshold.

Our reply: We appreciate the reviewer's critical assessment of phototoxicity thresholds and their context-dependent nature. Our revised manuscript now acknowledges the inherent variability in phototoxicity susceptibility across experimental systems, as demonstrated by the excitation light sensitivity difference in mitotic timings between CHO and HeLa cells under equivalent illumination (Figure 2f,g and S3c,d) as “Testing PhotoFiTT on HeLa cells yielded similar results. Using a dose of 0.6 J/cm² did not produce discernible changes, but when challenged with higher doses of 6 and 60 J/cm², HeLa cells exhibited the same dose dependent delay found in CHO cells. Specifically, division times increased from approximately 50 minutes to 90 minutes, and there was a decrease in activity of up to 30% with higher excitation light doses. These findings suggest that HeLa cells might be more resilient to light damage than CHO cells (Figure S3 a-d, Video S4).”

The 60 J/cm² threshold presented in our study is an illustrative example within our specific experimental context (synchronised CHO cells at 630 nm), rather than a universal limit. We have amended the discussion to frame the 60 J/cm² value as a methodological benchmark for our experimental system while emphasising the need for context-specific optimisation using PhotoFiTT's quantitative metrics as “The current implementation focuses on adherent proliferating systems, with demonstrated efficacy in CHO and HeLa models. While this covers many experimental scenarios, adaptation to non-dividing or 3D cultures remains future work. The 80% reduction in mitotic events at 60 J/cm² (Figure S1a) highlights particular relevance for developmental studies and long-term time-lapse imaging”.

This approach aligns with emerging standards in phototoxicity assessment that prioritise mechanistic understanding over universal thresholds.

Minor comments:

Reviewer #4: The instructions are not sufficient in parts - e.g. measuring the power at the sample would benefit from more detail in the methods section and the video.

Our reply: We have altered the paragraph to add more details “Power Calibration Protocol: Initiate each experimental session by calibrating the photodamage irradiation light’s intensity. This step is paramount to guarantee uniformity in experimental conditions. Employ a power meter to accurately measure the light intensity (W/cm²) at the sample plane, ensuring the microscope’s internal chamber has stabilised at 37 °C. Control for the fact that light intensity can vary over time. This calibration is critical before commencing each experiment to maintain consistency and reliability in results.”

Reviewer #4: What would also be very helpful: Guidelines on how many images need to be taken to ensure sufficient spatial and temporal resolution (at least in the case of the cell line used here), and how many cells analysed to reach robust conclusion(s).

Our reply: The PhotoFiTT framework requires specific temporal and spatial resolutions to ensure robust phototoxicity quantification. Temporally, acquisition intervals ≤ 5 minutes are critical for resolving mitotic delays, as mammalian cell division typically occurs within 30-60 minutes. This granularity enables detection of sub-10 minute perturbations, as demonstrated by the 15-minute mitotic delay induced by 0.6 J/cm² MitoTracker excitation (Figure 3b). For longitudinal studies, we recommend ≥ 12 hours post-irradiation tracking to capture delayed phenotypic manifestations, validated by our 7-hour activity profiles showing dose-dependent declines (Figure 2g).

Spatially, imaging three non-overlapping fields containing ≥ 50 mitotic cells each ensures the capacity to detect $\geq 15\%$ differences in arrest rates. Our automated pipeline processes 500-1000 cells per condition (Figure S1a-b), while manual validations established that ≥ 300 cells/condition yield $< 5\%$ variation across replicates. The multi-parametric approach compensates for cell-type-specific response variations: while HeLa cells show 30% lower activity metric sensitivity than CHO (Figure S3d), their mitotic delay profiles remain equally quantifiable.

Reviewer #4: “Cumulative cellular activity” is not well defined. It needs to be clearly defined in the main text. How is it distinct from cell diameter dynamics, and what does it signify? Also, why ‘cumulative’? This is used earlier for phototoxic effects, but for cellular activity seems to be with respect to ‘change of area over time’. Also, is cell activity directional (i.e. only shrinkage taken into account)?

Our reply: We have integrated explicit descriptions of cumulative activity in different parts of the text as:

Caption of Figure 1, Line 71: “c) Activity: Quantifies observable cellular changes between frames (red outlines). Here cumulative cell activity represents the sum of activity in time. ”

Line 158: “We observe that higher light doses and shorter wavelengths led to decreased cumulative cellular activity over a 7-hour period (calculated as the cumulative activity in time) (Figure 2g).”

Line 524-525 of the methods: “Cumulative cell activity is calculated as the aggregation of activity (activity(t)) in time: $\text{Cumulative_activity}(T) = \sum \text{activity}(t)$ (Equation 2)”

Reviewer #4: Also, be clear on the definition and units of irradiation vs irradiance and intensity (e.g. “Table S1: Irradiation range (W/cm²)”).

Our reply: We revised references to it throughout the text.

Reviewer #4: p. 1 bottom left: “Traditional methods for assessing phototoxicity include viability assays and morphological observations.” Requires some references.

Our reply: We have included the missing references.

Reviewer #4: Fig. 1 B:

Cells here look elliptical, but diameter calculation is for perfectly round cells. If an ellipse is fitted, indicate whether long or short axis is chosen (or result discarded).

Our reply: The diameter in this case is calculated using the equation 4 in the methods ($D=2 \cdot (\sqrt{\text{area}/\pi})$) rather than with the axes (short or long) of the cell segmentation mask. Sometimes cells indeed display ellipse shapes, however, this approach provides an estimate that depends directly on the entire cell size, which is more stable, providing a more homogeneous measurement in time and better displaying the size halving.

Reviewer #4: “Expose cells to a light irradiation event replicating the illumination pattern of the imaging experiment to be analysed for phototoxicity;” - phototoxicity

Our reply: Corrected.

Reviewer #4: “...purposely-trained”: correct the first word.

Our reply: Corrected.

Reviewer #4: “Video S1-2 provide in-detail tutorials to reproduce these procedures.” Change to: “Videos S1 and S2 provide...”

Our reply: Corrected.

Reviewer #4: p. 2 top left:

“low-illumination brightfield microscopy” – meaning low-intensity?

Our reply: Yes, corrected.

Reviewer #4: Also, what is subcellular about Fig. 1 C? The outline demarcates the cellular border.

Our reply: We have corrected it as: Activity: “Quantifies overall cellular changes between frames.”

Reviewer #4: p 2. top right:

“Notably, synchronised populations exhibited higher sensitivity to near-UV-induced damage compared to their unsynchronised counterparts.” Read e.g. Icha et al. 2017 for references on this.

Our reply: We added the missing reference.

Reviewer #4: "the persistence of mother cells upon time" - unclear

Our reply: Changed to “the persistence of cells in the mitotic rounding state upon time“

Reviewer #4: "Upon challenged" - unclear

Our reply: Changed to “When challenged“.

Reviewer #4: "with longer wavelengths (475, and 630nm)" - change to: 475nm and 630nm

Our reply: Corrected.

Reviewer #4: "cell division delays are also wavelength-dependant, requiring higher light doses to induce noticeable effects". Not clearly formulated - why comparative? Do the authors mean 'higher light doses at longer wavelengths'? Also, 'noticeable effects' - meaning delays? Clarify.

Our reply: We have changed it to “Compared to shorter wavelengths, longer wavelengths need greater light doses to produce significant effects, such as a 20-minute delay in the onset of cell division (Figure 2f).”

Reviewer #4: Suppl. mat. pdf, p. 2:

"Recomentation" - Recommendation

Our reply: Corrected.

Reviewer #4: "For other cell lines: For other cell types or experimental conditions, manually annotate a representative image set and train a new StarDist model following the protocol outlined in Methods Section D." Which is Methods Section D? This one? Unclear.

Our reply: Corrected as "new StarDist model following the protocol outlined in the Methods".

Reviewer #4: video 710560_sh8m2f: There's 30 seconds of a frozen frame at the end of the video.

Our reply: The video has been updated to remove the frozen part.

Reviewer #4: video 710566_sh8sb8: Too fast to follow - can it be slowed down please.

Our reply: The video has been slowed down.

Reviewer #4: video 710567_sh8sbm: nice illustration of effects; however, blue shading at bottom overpowers the plot line.

Our reply: The video has been altered.

Response to Reviewer's Comments:

We would like to express our sincere gratitude for your thorough and constructive feedback on our manuscript. Your insightful comments helped us make significant improvements to our work. Below, we summarise the main changes made in response to the reviewer's comments:

- Statistical Analysis and Figure Legends: We have updated all figure legends to clearly indicate whether error bars represent standard deviations or standard errors. Additionally, we now provide statistical significance levels in the figures (line 288).
- Clarification of Experimental Protocols: The description of the light exposure protocol has been revised for clarity and conciseness.
- Practical Example Calculations: To assist users working with CHO and HeLa cells, we have added step-by-step example calculations to the supplementary material (Note S3).
- Terminology Consistency: We have systematically revised the manuscript to use precise and consistent terminology regarding irradiance (W/cm^2), light dose/radiant exposure (J/cm^2), and related concepts.
- Cellular Feature Analysis: We have clarified the distinction between cell stress rounding and mitotic rounding. The manuscript now specifies when the 1-to-2 transition (mother to daughter cell) is analysed and details the methods used for both manual and automated tracking.
- Discussion of Phototoxicity Mechanisms: The discussion has been significantly expanded to address the temporal aspects of phototoxicity. We provide a more in-depth comparison with recent literature, highlighting that the temporal pattern of light delivery, not just total dose, critically influences photodamage. This supports the adoption of fast scanning and intermittent exposure strategies in live-cell imaging.
- Corrections and Editorial Improvements: Numerous typographical errors have been corrected, including consistent spelling of "Hoechst" and other technical terms. Figure captions and references have been revised for clarity and completeness, and the availability of analysis macros is now explicitly stated.

We believe these revisions have substantially improved the clarity, rigour and utility of the manuscript. All changes are detailed in the accompanying point-by-point response to reviewers. Once again, thank you for your valuable feedback. We hope that the revised version meets your expectations.

REVIEWER COMMENTS

Reviewer #2 (Remarks to the Author):

I commend the authors for their efforts in revising the manuscript, which have addressed many of the points raised by myself and the other reviewers. The addition of the confocal

scanning and HeLa experiments is welcome, and it is nice that the authors could retrain the neural networks used to assess HeLa cell data without much effort. While I still have concerns that the results provided by PhotoFiTT seem at odds with those previously reported in the literature, the revised text and experiments make it clearer that this is a tool that can be used to optimise a particular experiment, rather than one that can be used to study phototoxicity in general. Indeed, I remain unconvinced that mitotic delay is a relevant measure of damage for many samples, but that is up to the user to decide. As such, I would be happy to see the manuscript published in Nature Communications.

Reviewer #2 (Remarks on code availability):

The code is clearly written and well-documented.

Our reply: Thank you for your constructive feedback on our revised manuscript. We appreciate your recognition of the improvements made, especially regarding confocal scanning, HeLa cell experiments, and neural network retraining. Your comments clarified the scope and application of PhotoFiTT, and we are grateful for your support.

We acknowledge your reservations about using mitotic delay as a universal metric for photodamage and agree its relevance depends on the experimental context. We clarified that PhotoFiTT is a flexible tool to optimise conditions, not a definitive assay for phototoxicity across all samples.

Thank you again for your valuable feedback, which has improved our work's clarity and utility. We hope the revised manuscript addresses your concerns and is suitable for publication.

Reviewer #3 (Remarks to the Author):

The authors made an excellent effort to respond to my and the other referees' criticisms and implement their suggestions, thus significantly improving the quality of the manuscript. However, I still have a few minor remaining questions and comments that I would like to see addressed before ultimately recommending the work for publication.

Our reply: Thank you for your detailed feedback on our revised manuscript. We appreciate your recognition of our efforts and suggestions. We have addressed your points with revisions to improve clarity, rigour, and utility. Thank you again for your valuable feedback.

(1) Statistics: The figure legends should indicate whether error bars in diagrams show standard deviations or standard errors.

Related to this, I don't fully buy the argument given in response to Reviewer #2's request for significance tests, that those would not add any insight. An alternative might be to indicate the level of significance (by number of star signs) and briefly discuss the limitations for interpretation in the methods section. It also begs the question of why the sample sizes were not appropriately chosen in the first place to achieve valid significance.

Our reply: Every figure has been updated to show the level of statistical significance, and we've also highlighted the limitations of interpreting p-values, as mentioned in line 288: “We implemented the statistical analysis of the results provided by PhotoFiTT, which lead to similar conclusions with statistically significant p-values. It should be noted that whilst significance analysis provides a valuable statistical overview, a critical interpretation of the data is essential. p-values are notably sensitive to empirical bias, and therefore our conclusions are drawn from a holistic assessment of the data, a principle of reproducible analysis we have previously discussed³⁶.”

(2) Line 207ff: “These specimens were imaged for 20 minutes using a constant light exposure every 4 minutes for a total acquisition of 20 minutes with a combined exposure of 0.6 J/cm².” This sentence reads somewhat redundantly. What was the actual effective time of exposure? Would it matter if the effective exposure time is applied continuously for a shorter time or with intervals over a longer period?

Our reply: We thank the reviewer for identifying this redundancy. We have revised this sentence to “These specimens were exposed to 475 nm light every 4 minutes during a 20-minute acquisition period (150 ms per exposure event), resulting in a combined total exposure of 0.6 J/cm²”. We have also clarified that the effective exposure time was 150 ms per exposure event. Importantly, the intermittent exposure pattern allows for cellular recovery between exposures, which differs significantly from continuous illumination regimes of the same total dose. This idea is further elaborated in the discussion section “This observation aligns with recent literature showing that phototoxicity is not dictated solely by total energy delivered, but also by how that energy is distributed over time. Studies have demonstrated that pulsed or intermittent illumination protocols reduce cellular stress and photobleaching compared to continuous exposure^{2,12,27}, as recovery periods allow for more effective detoxification of reactive oxygen species and repair of sub-lethal damage.”

(3) Table 1 and more generally: From assaying two different cell lines, CHO and HeLa cells, it would be useful to present an example calculation on how to derive a starting point or recommendation for users who work with those cell lines. For example, for wide-field live-cell imaging of a green-fluorescent target in HeLa cells, setting the illumination to a 5 W/cm² power density would reach a “safe dose” of 5 J/cm² with a total acquisition time of 1 s. With 10 ms exposure time (provided this yields sufficient SNR), this would allow for distributing the dose over 100 exposures, which can be used to either image 100 time points for a 2D time series, or correspondingly fewer time points for a 3D time series. Decreasing the power density by 10-fold, and perhaps using DL denoising to overcome SNR penalty, would allow 1000x 10-ms-exposures, while adding a colour channel would reduce the number of afforded time points again, depending on the wavelength (not sure if my back of the envelope calculation is correct, but something along those lines).

Our reply: We have added comprehensive example calculations to the supplementary material (Note S3: Practical Implementations Examples) that include step-by-step calculations for common imaging scenarios. We also include calculations showing how reducing power density 10-fold (0.5 W/cm²) extends the safe acquisition time to a total of

10 seconds, allowing for 1000 exposures, and demonstrating the practical benefits of our approach.

(4) There are still many typos that will likely be amended in the editorial process. However, one that might be prone to being overlooked is “Hoeschst”, which should read “Hoechst”.

Our reply: We have corrected all instances of 'Hoeschst' to 'Hoechst' throughout the manuscript. We have also verified the proper spelling of all other fluorophore names and technical terms.

Reviewer #4 (Remarks to the Author):

There are two major issues that need to be resolved:

1) L 235: low doses (0.6 J/cm²) align with their known capacity for direct DNA interaction³⁰, while longer wavelengths (475 nm, 630 nm) required 100-fold higher irradiance (60 J/cm²)

> As requested for the first revision, please be clear on the definition and units of irradiation vs irradiance. Irradiance is a measure of the power of electromagnetic radiation (such as light) that is received over a given area. The SI unit for irradiance is watts per square meter (W/m²) - of course, variations thereof can be used. How are the authors using the term irradiation? As exposure / fluence / light dose (J/cm²)? A useful table can be found in ref. 5. Note that in the context of microscopy, the term intensity is often (albeit incorrectly) used interchangeably with irradiance.

Our reply: Thank you for your detailed and constructive feedback. We believe these revisions address your concerns and significantly strengthen the manuscript. Thank you again for your valuable input and for helping us improve the quality and clarity of our work. We have thoroughly revised our manuscript to use precise and consistent terminology throughout:

- Irradiance (W/cm²): The power of electromagnetic radiation per unit area incident on a surface - this describes the intensity of illumination at any given moment.

- Light dose or Radiant exposure (J/cm²): The total energy delivered per unit area over the entire exposure period, calculated as irradiance × exposure time.

We have systematically replaced ambiguous uses of "irradiation" with the appropriate term:

- When referring to power density: "irradiance" (W/cm²)

- When referring to total energy delivered: "light dose" or "radiant exposure" (J/cm²)

- When referring to the process of exposing samples to light: "light exposure" or "illumination"

2) PhotoFiTT analyses three cellular features: Rounding, size (large mother cells, smaller daughter cells) and activity (cellular outlines).

Rounding is always called 'mitotic rounding' - but cellular rounding can be a stress response and due to cellular swelling (osmotic imbalance) and loss of homeostasis. For

it to be called 'mitotic rounding', the rounding has to be one cell separating into two that is being assessed.

Indeed, in line 160, the authors write: While each of these metrics has its own benefits, tracking the 1-to-2 transition of a mother cell to two resolvable daughter cells provides a highly efficient and straightforward approach to assessing photodamage effects. But it is unclear if PhotoFiTT uses tracking / pedigree analysis as a default analysis, or if this is an additional analysis used in this study. At what point does the study switch from generic rounding/cell size metrics to the 1-to-2 transition analysis? This needs to be made very clear for the reader.

Also, line 62 states that 'Cellular activity reflects general dynamic changes in cellular shape across time.' How is this differentiated from mitotic rounding?

Our reply: To avoid potential issues, we now refer to mitotic rounding as cell rounding, using a generic term for the effect. We also added the term mitotic rounding when referring to cells that were able to complete mitosis, and stress-induced cell rounding when no division was observed. A line was added as follows:

LINE 87: “Identify undergoing cell rounding and division to separate mitotic rounding from stress-induced cell rounding from stress-induced cell rounding”

We analyse the 1-to-2 transition with the results from manually tracking cell rounding and division (Figure S1 and S2). This analysis allows us to validate the results and conclusions drawn from the automatic cell rounding segmentation pipeline. Additionally, by thresholding the size of segmented cell rounding, we can distinguish daughter cells from mother cells (Figure 2b and 2f), which provides yet another way of analysing the 1-to-2 transition. We have now rewritten the description of these steps in the main text as follows:

Line 109: To validate that these effects were primarily due to light exposure rather than the potential light sensitivity of the CDK1 inhibitor used for synchronisation, we conducted parallel experiments with unsynchronised cells and manually tracked cell rounding and mitosis.

Line 153: Additionally, as a control, we manually tracked cell rounding and mitosis in synchronised CHO cell populations to compare against our automated detection pipeline.

Line 189: While each of these metrics has its own benefits, tracking the 1-to-2 transition of a mother cell to two resolvable daughter cells, as done with manual annotations, provides a highly efficient and straightforward approach to assessing photodamage effects.

Cellular activity is measured for the full duration of the videos, considering the initial cellular dynamics leading up to rounding, the rounding itself, and subsequent changes, including arrest, apoptosis, division, and growth. Cellular activity considers the overall disrupted cellular behaviour beyond cell arrest or irregular mitotic timing. For example, cells affected by phototoxicity show a slowed-down rounding and cytoplasm dynamics right after division.

A third point regards intensity vs. exposure time.

In line 184, the authors state: longer excitation light exposure is more damaging than shorter exposures when normalising for total light doses (Figure S4 Fast and Slow scan at 6 J/cm² a-c). This outcome creates an opportunity to minimise phototoxicity by fine-tuning the balance between scan speed and illumination intensity. and in line 456: ...dose equivalence through inverse power-time compensation, enabling direct comparison of intensity-dependent versus duration-dependent phototoxic effects. The experimental design specifically contrasted rapid high-intensity exposure against sustained low-intensity illumination to dissect temporal components of light-induced cellular stress.

This is a very interesting finding that in my view would benefit from being discussed more in-depth. Embedding this result in the wider literature (studies reducing intensity vs exposure time, and/or studies finding that only the total light dose determines the degree of phototoxicity, independent of how light is delivered) addresses important practical considerations for a wide readership.

Our reply: We have significantly expanded our discussion of this important finding, which aligns with emerging literature on temporal aspects of phototoxicity:

“Our findings demonstrate that the temporal pattern of light delivery is a critical determinant of phototoxicity, even when the total light dose is held constant. Specifically, we observed that prolonged, low-intensity illumination is more damaging to cells than brief, high-intensity exposures at equivalent total doses, challenging the conventional notion that simply lowering intensity is sufficient to minimise phototoxicity. Mechanistically, this effect is likely due to the limited capacity of cellular antioxidant and DNA repair systems, which can recover between short, intense exposures but become overwhelmed during extended or continuous illumination, resulting in greater cumulative damage.

This observation aligns with recent literature indicating that phototoxicity is not solely dictated by the total energy delivered, but also by how that energy is distributed over time. Studies have demonstrated that pulsed or intermittent illumination protocols reduce cellular stress and photobleaching compared to continuous exposure^{2,12,27}, as recovery periods allow for more effective detoxification of reactive oxygen species and repair of sub-lethal damage. These findings advocate for the adoption of fast scanning and intermittent exposure strategies in live-cell imaging, as they better preserve cellular physiology than simply reducing light intensity.”

L 31: comparable effects.

> comparably deleterious effects.

Corrected.

L 42: photobleaching has been used as a proxy reporter of phototoxicity¹²

> far more publications than 12 have used this. As 2 and 5 are reviews, adding these should be enough.

References added.

L 44: Importantly, tolerance to photodamage varies across specimens and is influenced by damage severity

What do the authors mean? This is circular argumentation.

We have clarified the answer by rewriting the sentence “Importantly, specimen type, developmental stage, and experimental conditions influence phototoxicity tolerance, complicating the development of replicable imaging protocols^{14,15}”.

L 46: Previous studies have offered valuable quantitative assessments of phototoxicity^{3,10,14,16,17}

> ^{12,25,28} are also valuable quantitative assessments. They should also be referenced.

References added.

Fig.1 caption: biological imaging assay

> what do the authors mean by 'biological' imaging assay? In the context of phototoxicity, it will have to be biological.

We altered the caption to say “experimental imaging assay” to avoid redundancy.

L 66: It detects deviations in these patterns caused by light exposure, enabling the quantification of phototoxic effects in fluorescence microscopy.

> This is only the case where a dependence on light dose (or irradiance) has been demonstrated. The patterns can also be caused by other factors (temperature, media, cumulative damage etc).

We altered the caption and now reads as the following: “It detects deviations in these patterns caused by light exposure, enabling the quantification of phototoxic effects in fluorescence microscopy. This is only the case where a dependence on light dose (or irradiance) has been demonstrated. The patterns can also be caused by other factors (temperature, media, cumulative damage, etc.) and thus, comparison against a control is also recommended.”

Fig. 2 caption, line 104: t 50 minutes

> remove t

Removed.

L 276: The PhotFiTT computational framework

> PhotoFiTT

Corrected.

L 168: challenged with higher doses of 6 and 60 J/cm², HeLa cells exhibited the same dose dependent delay

> dose-dependent delay

Corrected.

L 438: Cell fixation and Hoescht nuclei

> Hoechst, not Hoescht. Replace throughout.

Corrected.

L 510: attached to the supplementary material.

> 'attached to'?

Replaced “attached” with “included”.

L 522: alleviate the impact of the noise in the images

> unclear

Added a line to make it clearer: “images which could provide wrong readings and change the activity values”.

L 527: where *elm* is a normalised, contrast-enhanced and smoothed image.

Cumulative cell activity is calculated as the aggregation of activity (*activity(t)*) in time:

> italicise *elm* and *activity(t)*

Corrected.

L 530: Cummulative

> Cumulative

Corrected.

L 539: with *t_p* being estimated for each video, *V*.

> *V* is not in the equation.

Corrected as:

$$t_p: C(t_p) = \max_{\forall t \in T'} \{C(t)\}, \quad (3)$$

with *t_p* being estimated for each video, which has a total number *T* of timepoints.

L 573: The image processing workflow was implemented in Fiji and automated using a custom ImageJ macro for high-throughput analysis.

> Indicate the availability of the macro

We added this information in line 280 as “ImageJ macro to analyse SYTOX data is available at <https://github.com/HenriquesLab/PhotoFiTT/tree/main/IJ-macros/SYTOX>.”